# THE VISUAL TASK ADAPTATION BENCHMARK

## ABSTRACT

Representation learning promises to unlock deep learning for the long tail of vision tasks without expansive labelled datasets. Yet, the absence of a unified yardstick to evaluate general visual representations hinders progress. Many sub-fields promise representations, but each has different evaluation protocols that are either too constrained (linear classification), limited in scope (ImageNet, CIFAR, Pascal-VOC), or only loosely related to representation quality (generation). We present the Visual Task Adaptation Benchmark (VTAB): a diverse, realistic, and challenging benchmark to evaluate representations. VTAB embodies one principle: good representations adapt to *unseen* tasks with *few examples*. We run a large VTAB study of popular algorithms, answering questions like: How effective are ImageNet representation on non-standard datasets? Are generative models competitive? Is self-supervision useful if one already has labels?

## 1 INTRODUCTION

Deep learning has revolutionized computer vision. Distributed representations learned from raw pixels have enabled unprecedented performance on many visual understanding tasks. Hand-crafted features have been replaced with hand-annotated datasets, with thousands to millions of examples (Krizhevsky, 2009; Russakovsky et al., 2015). By contrast, humans learn a wide range vision tasks using just a few examples per task. A key research challenge is to close this gap in sample efficiency, and unlock deep learning for the long tail of problems without many labels.

The goal of improving sample efficiency has been approached from many angles, such as: few-shot learning (Fei-Fei et al., 2006), transfer learning (Pan & Yang, 2009), domain adaptation (Wang & Deng, 2018), and representation learning (Bengio et al., 2013). Representation learning is studied in many contexts: supervised pre-training (Sharif Razavian et al., 2014), self-supervised learning (Doersch et al., 2015), semi-supervised learning (Chapelle et al., 2009), generative modeling (Donahue et al., 2017), and disentanglement learning (Higgins et al., 2017). Yet many questions remain unanswered, such as: Do representations trained using ImageNet labels generalize to diverse tasks? Does the great progress on generative modeling translate in better general representations? Is self-supervision useful when labelled datasets already exist? Are semantic image-level labels useful for tasks that require understanding the structure of the world?

The lack of a common benchmark impedes the understanding of visual representations; each subdomain has its own evaluation protocol, and it is hard to assess different approaches. Benchmarks have been critical in other domains, such as RL (Mnih et al., 2013), natural image classification (Russakovsky et al., 2015), and NLP (Wang et al., 2018). Inspired by these successes, we propose a benchmark that follows similar principles: (i) we put minimal constraints to encourage creativity, (ii) focus on practical considerations, and (iii) make it challenging.

We present the Visual Task Adaptation Benchmark (VTAB). VTAB is based on a single principle: a better algorithm is one that solves a diverse set *previously unseen* tasks with *fewest possible labels*. The focus on sample complexity reflects our belief that learning with few labels is the key objective of representation learning. Task diversity is also crucial to assess a representation's generality. Therefore, our benchmark extends beyond standard natural classification tasks, and includes those related to sensorimotor control, medical imaging, and visual reasoning.

We treat the benchmark tasks as *unseen*, but otherwise, we do not constrain the representation learner, or transfer algorithm. Pre-training may be performed on any data (labelled or unlabelled), provided that it does not overlap with the benchmark tasks, which would violate the notion of *unseen*.

To transfer the representations one may freeze, fine-tune, add new layers, apply domain adaptation, meta-learning, or any other technique.

With VTAB, we perform an extensive representation study, which reveals: (i) Supervised pre-training on ImageNet yields useful representations for natural images. Surprisingly, these work on distant domains, but to a lesser extent. (ii) Self-supervision is effective, although other unsupervised methods, such as generative models appear less so. (iii) Combining self-supervised with supervision outperforms all other studied algorithms, even without additional unlabelled data. (iv) Restricted evaluation using a linear classifier could lead to different conclusions. Data, code, continuously updated results, and models used are made available at [`anonymous-url`].

## 2  THE VISUAL TASK ADAPTATION BENCHMARK

We seek algorithms that perform well on a wide variety of unseen visual understanding tasks with few labels per task.[1] We first formalize this objective and then specify a practical benchmarking procedure to measure progress.

A *dataset* $D^n$ is a set of $n$ instances $\{(\boldsymbol{x}_i, \boldsymbol{y}_i)\}_{i=1}^n$ with observations $\boldsymbol{x}_i \in X$ and labels $\boldsymbol{y}_i \in Y$. A *prediction function* is any mapping $F : X \to Y$ (e.g. a classifier). A (learning) *algorithm*, $\mathcal{A}$, takes as input a dataset and outputs a prediction function. For example, $\mathcal{A}$ may be a pre-trained neural network coupled with a training mechanism. An *evaluation procedure*, $\mathcal{E}_T$, takes $F$ and outputs a scalar measuring $F$'s performance (e.g. test-set accuracy).

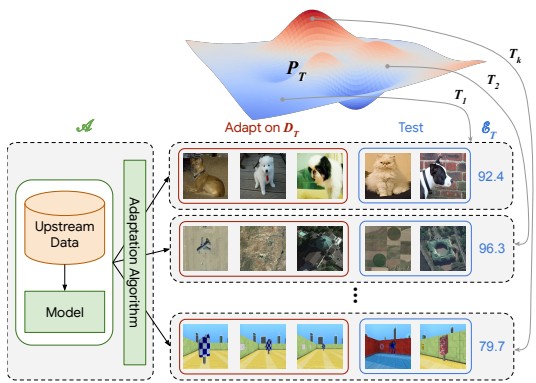

Figure 1: Overview of the VTAB protocol.

We seek the algorithm that maximizes the expected performance over a distribution of tasks $P_T$, where a task $T$ is a tuple containing a task-specific dataset distribution $D_T$ and evaluation procedure. Given access to only $n$ samples per task we aim to maximize:

$$\text{SCORE}_n(\mathcal{A}) = \mathbb{E}_{T \sim P_T} \, \mathcal{E}_T \left[ \mathcal{A}(D_T^n) \right], \tag{1}$$

This general formulation requires a few clarifications. First, the task distribution needs to be appropriate; we desire a spectrum of tasks covering those solvable by a vision algorithm with human-like capabilities. Second, $n$ may be varied to measure the relative sample complexity of different algorithms. In practice we choose it to be similar to that of a modest labelling budget (Section 3.1). Third, we assume that $P_T$ is known, and the goal is to build an algorithm with the best inductive biases for solving samples from it. We desire an "open world", where evaluation data is extremely diverse and can always be freshly sampled. Unfortunately, in practice the benchmark must contain a finite number of fixed test sets, therefore we must ensure that the algorithms have not been pre-exposed to specific evaluation samples, as described below.

### 2.1  A PRACTICAL BENCHMARK

We now describe the Visual Task Adaptation Benchmark (VTAB), designed to be the best possible proxy for Eq. (1). Fig. 1 contains an overview diagram. We then discuss a suite of methods that may be used to optimize VTAB: deep representation learning.

Eq. (1) is very difficult to compute in practice. One must define a representative $P_T$ and compute the desired expectation. If sampling the distribution, one must avoid overfitting to the sample. Finally, limiting computational complexity becomes paramount as the number of tasks grows.

**Distribution over tasks**  We aim for universal visual understanding, so we informally define $P_T$ as "Tasks that a human can solve, from visual input alone.". See Appendix B where we validate this. Intuitively, such tasks should benefit from visual representations learned by observing and interacting with the natural world. The second clause eliminates tasks that require external knowledge that

---

[1]In some practical settings additional *unlabelled* data may be available for unseen tasks in addition to the labelled data. We omit this setting, leaving it to future work.

is (currently) unreasonable for a vision algorithm to acquire – an extreme example would classifying objects grouped by a function of their spelling in a natural language. Section 2.2 details the samples.

**Expectation over tasks**  We approximate the expectation over tasks by an empirical average over a number of hand-picked samples. Ideally, we would sample a new task for each evaluation, as is possible in procedural environments, e.g. (Finn et al., 2017). However, real-world vision tasks are expensive to collect, so we define a fixed, representative set of samples from $P_T$. While fixing the set reduces variance, it introduces a risk of meta-overfitting.

**Mitigating meta-overfitting**  To do this, we treat the evaluation tasks like a regular test set, and thus consider them unseen. This implies that algorithms that use pre-training must not pre-train on any of the evaluation tasks (even their unlabelled images). Upstream training should provide useful inductive biases for any draw from $P_T$, so should not use test samples. Fortunately, despite numerous test re-evaluations on popular ML benchmarks, such as ImageNet, progress seems to transfer to new data (Recht et al., 2018; 2019; Kornblith et al., 2019).

**Unified implementation**  We use many diverse tasks, however, this could render usage impractical. As discussed, the algorithms must have no prior knowledge of the downstream tasks. Therefore, while it is permitted to run a hyperparameter search on each task, the search space cannot be task-dependent. To get meaningful results, we need to define architectures and hyperparameter searches that work well across the benchmark.

For this, we convert the tasks into classification problems. With a homogeneous task interface, we may control for possible confounding factors. For example, we may use the same architecture and hyperparameter sweep throughout. Of course, not all tasks can be efficiently modeled as classification with global labels, such as those requiring per-pixel predictions. Nonetheless, we design the tasks such that success on VTAB requires learning of diverse set of visual features: object identification, scene classification, pathology detection, counting, localization, 3D geometry, and others.

## 2.2 TASKS

VTAB contains 19 tasks which cover a broad spectrum of domains and semantics. Appendix A contains details. These are grouped into three sets: • NATURAL, • SPECIALIZED, and • STRUCTURED.

The • NATURAL group represents classical vision problems. These tasks contain natural images captured using standard cameras. The classes may represent generic, fine-grained, or abstract objects. The group includes: Caltech101, CIFAR-100, DTD, Flowers102, Pets, Sun397, and SVHN.

The • SPECIALIZED group also contains images of the world, but captured through specialist equipment. These images contain invariances from those in the • NATURAL tasks. Nonetheless, humans recognize the structures therein, thus generic visual representation should also capture the visual concepts in these images. We have two sub-groups: remote sensing, and medical. Remote sensing includes Resisc45 and EuroSAT: aerial images of the earth captured using satellites or aerial photography. Medical includes Patch Camelyon, metastases detection from microscopy images, and Diabetic Retinopathy, retinopathy classification from fundus images.

The • STRUCTURED group assesses comprehension of the structure of a scene, for example, object counting, or 3D depth prediction. Most of these tasks are generated from simulated environments, whose structure is easy for a human to determine, but whose domain differs greatly to datasets like ImageNet. These tasks are intended as a step towards useful representations for perceptual control. We include: *Clevr*: Simple shapes rendered in a 3D scene, we include two tasks: counting and depth prediction. *dSprites*: Simple black/white shapes rendered in 2D, we include two tasks: location and orientation prediction. *SmallNORB*: Artificial objects viewed under varying conditions, we include two tasks: object-azimuth and camera-elevation prediction. *DMLab*: Frames from a rendered 3D maze. The task involves predicting the time for a pre-trained RL agent to navigate to an object. *KITTI*: frames captured from a car driver's perspective. We predict the depth of the nearest vehicle.

## 2.3 REPRESENTATION AND TRANSFER LEARNING

Success on VTAB, and ultimately optimizing Eq. (1), requires some knowledge of $P_T$. For example, algorithms that use CNN architectures have a useful inductive bias for vision. While these biases are usually manually designed (e.g. through choice of an architecture), they can also be learned.

For example: (i) Zoph & Le (2017) learn performant architectures, (ii) Bello et al. (2017) learn optimization algorithms, (iii) Wong et al. (2018) learn hyperparameter search strategies, (iv) Cubuk et al. (2019) learn data augmentation functions, (v) Raghu et al. (2019) learn layer-wise initialization distributions, and (vi) pre-trained data representations can be learned.

While all of the above merit investigation for VTAB, we focus on representation learning. Human visual perception is refined through years of observation and interaction with the world, resulting in a system that solves new tasks with few in-domain labels. Likewise, we aim to pre-train a network that extracts useful features, or representations, from raw data.

**Transfer strategy**  When representations are not pre-trained on the evaluation tasks themselves (which is explicitly forbidden in VTAB), they must be adapted to solve the new task (using limited in-domain data). State-of-the-art representation learning algorithms typically train a deep neural network on some task(s), and then adapt the network to the new task. The simplest adaptation strategy is to *freeze* the network's weights and train another – usually smaller – model on top. When the upstream and downstream datasets differ significantly, *fine-tuning* the original weights is more effective (Yosinski et al., 2014; Kornblith et al., 2019). VTAB does not constrain the transfer strategy; here we benchmark methods using fine-tuning since it tends to perform best.

**Upstream training**  The representation learning literature often focuses on unsupervised learning as it may be applied to any data without human annotation. However, supervised data, where available, may also produce good representations. Indeed, the most popular models used in practice are pre-trained on ImageNet labels (Huh et al., 2016, and refs therein). VTAB *does not constrain the type of data* (supervised or unsupervised) used to pre-train the representations.

## 3 EXPERIMENTS

We use VTAB to evaluate popular representation learning algorithms. We ask four primary questions: (i) How effective are these algorithms, such as ImageNet pre-training, across a broad spectrum of tasks? (ii) How do different algorithm classes compare? (iii) Which method achieves the best overall performance on VTAB? (iv) How does VTAB compare to other benchmarking strategies?

### 3.1 SETUP

**Data**  Section 2.2 describes the tasks. VTAB aims to assess adaptation with limited data, so we evaluate primarily on 1,000 training examples per task, drawn i.i.d. from each training set. This simulates performance under a small per-task labelling budget which is the main goal of this benchmark. For completeness, we also use the full datasets. This allows us to assess the value of representation learning as in-domain data increases, and to check performance against prior art.

**Representation learning algorithms**  We evaluate the following representation learning algorithms: supervised, semi-supervised, self-supervised, and generative models. As a baseline we train a "from-scratch" model, starting from random weights. We evaluate a total of 16 algorithms. Throughout, the models are pre-trained ImageNet.[2]

The supervised model is trained on 100% of the ImageNet labels (SUP-100%). For self-supervised learning, we include both image-based and patch-based models. The image-based models include ROTATION (Gidaris et al., 2018) and EXEMPLAR (Dosovitskiy et al., 2014). The patch-based models include RELATIVE PATCH LOCATION (Doersch et al., 2015) and JIGSAW (Noroozi & Favaro, 2016). We use the public implementations of Kolesnikov et al. (2019). The patch-based models are converted into image-level classifiers by averaging the representations from a $3 \times 3$ grid of patches (see Appendix C). The semi-supervised models are trained using 10% of labelled ImageNet data with an auxiliary loss on all of the data, see (Zhai et al., 2019). We use either rotation (SUP-ROTATION-10%) or Exemplar (SUP-EXEMPLAR-10%) auxiliary losses. These models can also use all of the labels, denoted SUP-ROTATION-100% and SUP-EXEMPLAR-100%. For generative models, we use GANs and VAEs. The most common way to extract representations from a GAN is from the discriminator, replacing its final linear layer. We use both the label-conditional and unconditional BigGAN discriminators (Brock et al., 2019) (COND-BIGGAN and UNCOND-BIGGAN) using the public implementations of Chen et al. (2019); Lucic et al. (2019). For autoencoders, the output of

---

[2]ImageNet is not a VTAB evaluation task by design to permit upstream training on this dataset.

the encoder can be used as the representation. We test VAEs (Kingma & Welling, 2014), and WAEs with three distribution matching losses: GAN, MMD (Tolstikhin et al., 2018), and UKL (Rubenstein et al., 2019).

**Transfer algorithm**   VTAB permits any transfer algorithm. We use fine-tuning of the entire network here, since that performs best (Kornblith et al., 2019). Adding a linear model to frozen networks is also popular, so we compare to this approach in Section 3.4.

**Hyperparameters**   Both the upstream pre-training of representations and transfer involve many hyperparameters. One major choice is architecture. Bigger architectures almost always perform better (see Appendix L and Kolesnikov et al. (2019)). Therefore, for comparability, we use ResNet50-v2, or similar, for all methods. For the supervised and semi-supervised methods we use the architecture in He et al. (2016). For GANs and auto-encoders we also use deep residual networks, but with appropriate modifications to train them successfully on their generative pre-training task. Appendix C contains details.

The hyperparameters used for transfer also influence performance, and different datasets may require different settings. We run VTAB in two modes: *lightweight* and *heavyweight*.

The lightweight mode performs a restricted hyperparameter search on each tasks, permitting fair comparison with few resources. Many hyperparameters are fixed, including optimizer, batch size, pre-processing, and weight decay. For each task, lightweight mode sweeps four hyperparameters: 2 initial learning rates, and 2 learning rate schedules. See Appendix I for details. We choose short schedules to limit cost, but show in Section 3.5 that these yield near-optimal performance.

In heavyweight mode we perform a large hyperparameter search, limited only by resource availability. Here, we perform a random search over learning rates, schedules, optimizers, batch size, train pre-processing functions, evaluation pre-processing functions, and weight decay. We include longer training schedules and higher resolution images. This mode is used to establish the best attainable VTAB score with the algorithms studied. Details in Appendix I.

Due to the high cost of heavyweight mode, we perform our main study in lightweight mode. In Section 3.3 we show that although extensive tuning improves all methods, the relative performances are mostly unaffected. For future study using VTAB, we recommend a similar lightweight setup that facilitates fair comparison without high cost. If large computation resources are available, the heavyweight mode may be used to push the state-of-the-art.

**Protocol**   We perform model selection for each task on a validation set. We then re-train the best model on the training and validation set (or a 1000-example subset thereof), and evaluate on the test set. We run the test set evaluation three times and report the median score by default.

**Metrics**   We evaluate all methods with top-1 accuracy. We consider other reported metrics in Appendix D. To aggregate scores across methods, we compute the mean top-1 accuracy. We investigate more complex aggregation strategies in Section 3.5, but the relative performances remain the same, so we report mean accuracy due to its simplicity.

## 3.2   VTAB RESULTS

We run all 16 methods on 19 tasks with 1000 training examples and the full datasets using lightweight mode. We assess the value of representation learning over from-scratch training, and compare the performance of the different methods across the dataset groups. Fig. 4 shows two detailed per-task comparisons: (i) SUP-100% (ImageNet) versus FROM-SCRATCH which shows the value of standard ImageNet representations. (ii) SUP-ROTATION-100% versus SUP-100% which shows the best performing method versus ImageNet. Appendix G, Table 6 contains the complete results table.

**Generative models**   These perform worst, often more poorly than training from scratch. This is perhaps unsurprising because scaling these models to complex datasets is known to be challenging. Nonetheless, despite SOTA generative quality on ImageNet, the GAN's discriminator does not provide useful representations. On the 1000-example subsets the UNCOND-BIGGAN outperforms from-scratch, but all generative models lose substantially on the full datasets. Interestingly, on the full dataset the GANs perform better than autoencoders on the ● NATURAL tasks, but worse on ● STRUCTURED. This indicates that they may be more strongly fit to the ImageNet domain.

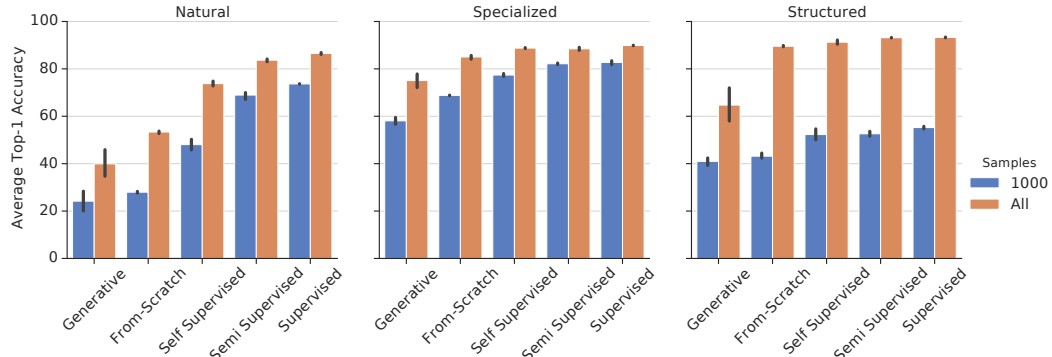

Figure 2: The methods are divided into five groups: Supervised with $100\%$ labels, semi-supervised with $10\%$ labels, self-supervised without labels, generative models, and training from-scratch. The plots show results on ● NATURAL, ● SPECIALIZED, and ● STRUCTURED datasets, respectively. The bar height indicates the mean top-1 accuracy across all methods and 3 test repeats within each group. The error bars indicate the variance across methods and runs.

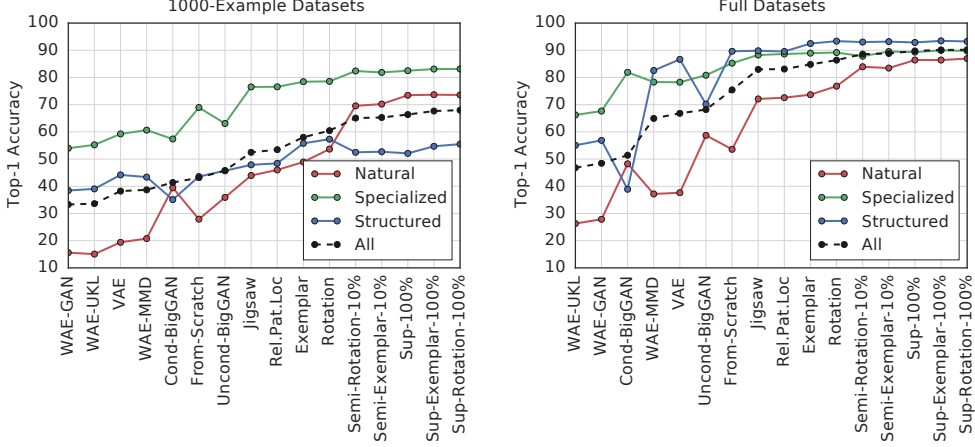

Figure 3: Average top-1 accuracy across the tasks in each group. The x-axis indexes the methods. The methods are ordered according to their average accuracy across all datasets (dashed curve).

Similarly, Ravuri & Vinyals (2019) show that models with high sample quality fail to generate data from which an accurate classifier can be trained. Explicit consideration of representation quality, in addition to generative scores, may lead to further progress in generative modeling research.

**Self-supervised** All self-supervised representations outperform from-scratch training. The best, ROTATION attains $60.4\%$ mean top-1 accuracy, while FROM-SCRATCH attains $43.1\%$ (1000-example datasets). Methods applied to the entire image (ROTATION, EXEMPLAR) outperform patch-based methods (JIGSAW, REL.PAT.LOC.). However, the patch based methods perform slightly better on DTD (Describable Textures) and Retinopathy (Fundus images), see Appendix G, Table 6. These tasks require sensitivity to local textures, which is well represented by patch-based methods. Figure 2 shows that self-supervised perform worse than supervised methods on the ● NATURAL datasets, but similarly on ● SPECIALIZED, and even better on some ● STRUCTURED tasks. This indicates that ImageNet labels may encode sub-optimal invariances for non-natural tasks.

**(Semi-)Supervised** Supervised models perform best. The benefits are most pronounced on the ● NATURAL tasks, whose domain and task-semantics are arguably most similar ImageNet classification. The combination of self-supervision with $10\%$ labels (semi-supervised) performs only marginally worse $(1 - 1.5\%)$ than methods with $100\%$ labels. Interestingly, there is benefit of com-

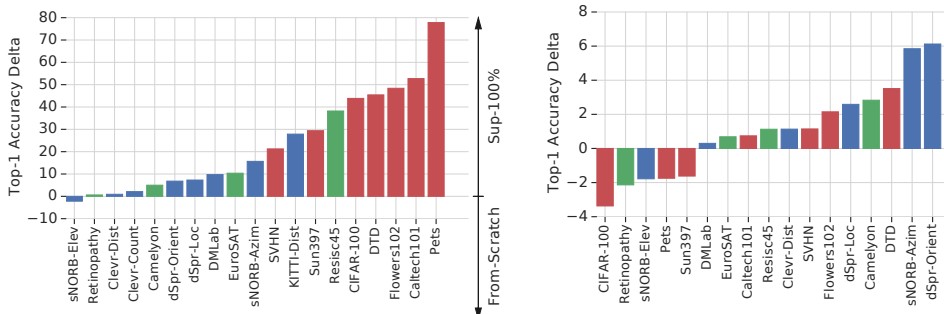

Figure 4: Absolute difference in top-1 accuracy between method pairs for each dataset. The bar colour indicates the task group: ●NATURAL, ●SPECIALIZED, and ●STRUCTURED. Left: SUP-100% versus FROM-SCRATCH — supervised pre-training yields a large improvement on the ●NATURAL datasets and some others. Right: SUP-ROTATION-100% versus SUP-100% — the additional self-supervised loss on top of supervised loss yields improvements, especially on the ●STRUCTURED tasks. Note that the y-scales differ; the left-hand side's differences are much larger.

| | | Caltech101 | CIFAR-100 | DTD | Flowers102 | Pets | SVHN | Sun397 | Camelyon | EuroSAT | Resisc45 | Retinopathy | Clevr-Count | Clevr-Dist | DMLab | KITTI-Dist | dSpr-Loc | dSpr-Ori | sNORB-Azim | sNORB-Elev | Mean |
|---|---|---|---|---|---|---|---|---|---|---|---|---|---|---|---|---|---|---|---|---|---|
| **1000** | From-Scratch | 60.2 | 24.5 | 51.1 | 73.4 | 43.1 | 88.3 | 14.9 | 82.1 | 93.3 | 63.9 | 75.4 | 53.0 | 57.3 | 36.6 | 55.7 | 88.6 | 67.6 | 52.2 | **81.6** | 61.2 |
| | Sup-100% | 89.7 | **52.7** | 65.4 | 94.2 | 90.1 | 91.0 | **35.5** | **87.3** | 96.0 | 84.0 | 78.0 | 59.0 | **63.0** | 44.4 | **80.1** | **89.3** | 70.7 | 57.5 | 54.7 | 72.8 |
| | Sup-Rot-100% | 88.0 | 48.9 | **68.1** | **94.7** | 89.0 | **91.8** | 33.7 | 86.2 | **96.6** | **85.2** | **78.5** | 66.5 | 61.5 | **45.5** | 77.2 | 88.8 | **71.8** | 59.4 | 52.0 | 72.8 |
| | Sup-Ex-100% | **90.4** | 50.4 | 66.1 | 94.3 | **90.6** | 91.3 | 35.2 | 86.7 | 95.8 | 83.3 | 78.4 | **73.8** | 62.4 | 44.1 | 79.7 | 87.5 | 71.1 | **62.8** | 54.8 | **73.6** |
| **Full** | From-Scratch | 74.5 | 77.8 | 67.1 | 85.8 | 70.9 | 97.0 | 70.1 | **91.2** | 98.8 | 94.3 | 82.8 | 99.8 | 96.7 | 76.6 | 68.4 | **100.0** | 96.7 | 99.9 | 94.0 | 86.4 |
| | Sup-100% | 93.0 | 83.4 | 73.7 | 97.3 | 92.7 | 97.5 | 75.6 | 87.3 | 98.8 | 96.1 | 83.4 | 100.0 | **97.0** | 78.8 | 81.0 | **100.0** | 96.8 | **100.0** | 98.5 | 91.1 |
| | Sup-Rot-100% | 93.5 | **84.0** | 76.8 | 97.4 | 92.6 | 97.4 | **75.6** | 86.5 | **99.1** | 96.3 | **83.7** | 100.0 | 96.8 | 79.6 | 82.4 | **100.0** | 96.8 | **100.0** | 96.7 | 91.3 |
| | Sup-Ex-100% | **93.8** | 83.1 | 76.5 | **97.8** | **92.9** | **97.5** | 75.3 | 86.5 | 99.0 | 96.3 | 83.7 | 99.9 | 96.8 | 79.3 | **82.8** | **100.0** | 96.7 | **100.0** | **99.1** | **91.4** |

Table 1: Top-1 accuracy of the top performing models and Resnet50 from-scratch.

bining self-supervised losses with 100% labelled ImageNet, i.e. without additional unlabelled data, see Fig. 4. SUP-ROTATION-100% attains 66.4%, whereas SUP-ROTATION-100% attains 68.0%.

### 3.3 HEAVYWEIGHT HYPERPARAMETER SWEEPS

We compare FROM-SCRATCH and the three best models: SUP-100%, SUP-ROTATION-100%, and SUP-EXEMPLAR-100% using the heavyweight hyperparameter search. Table 1 contains the results. The hyperparameters selected are shown in Appendix I. As expected, all methods improve significantly, e.g. FROM-SCRATCH improves from 43.1% to 61.2% (1000-examples). Prior work (He et al., 2018) shows that with sufficient data and training steps, from-scratch training is competitive for detection. We observe, with the heavy sweep, that although the gap to SUP-100% is smaller, it is still substantial: SUP-100% attains 72.8%. Again, addition of self-supervision helps, but the margin is smaller. SUP-EXEMPLAR-100% performs best with a VTAB (1000-example) score of 73.6.

We check our results against those recently reported in the literautre (Appendix H, Table 8). Our results are comparable; behind on highly popular tasks on which more complex architectures have been applied (e.g. CIFAR), but ahead in others. In Appendix L we show that simply increasing the architecture size improves VTAB performance.

In summary, pre-trained representations still offer substantial value with heavy hyperparameter tuning and long training schedules on VTAB. The overall ranking of methods remains the same, with the exception that SUP-EXEMPLAR-100% now outperforms SUP-ROTATION-100%, although these methods' performances are always similar.

## 3.4 FROZEN FEATURES EXTRACTORS

Representations are often evaluated by training a linear layer on a frozen model (Kolesnikov et al., 2019). ImageNet is often used for linear evaluation, but this is meaningless for the models we consider because many use ImageNet labels for pre-training. However, linear evaluation is also used in a transfer setting (Goyal et al., 2019a), so we apply this protocol to the VTAB tasks, and contrast to fine-tuning used above. The full protocol resembles the lightweight fine-tuning sweep, see Appendix K.

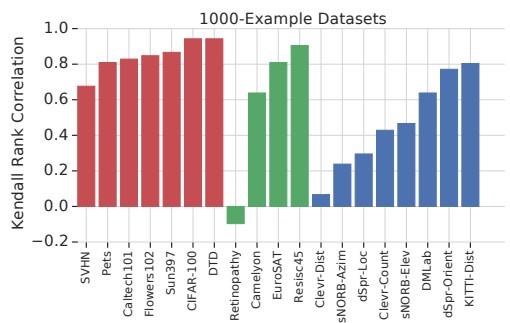

Figure 5: Kendall rank correlation coefficient between fine-tuning and linear evaluation on each dataset.

Fig. 5 shows the per-task correlation between linear and fine-tuning. Appendix K, Table 9 contains the full table of results. We first note that linear evaluation significantly lowers performance, even when downstream data is limited to 1000 examples. For example, SUP-100% attains 66.4% with fine-tuning (lightweight sweep), but 57.3% with linear. Linear transfer would not by used in practice unless infrastructural constraints required it. Second, Fig. 5 shows that on many datasets, particularly ● SPECIALIZED and ● STRUCTURED, the correlation is low. Linear evaluation may lead to different conclusions, for example, COND-BIGGAN attains 43.3% (1000-examples) on linear, outperforming both both patch-based self-supervised methods (REL.PAT.LOC and JIGSAW). Yet when fine-tuned these patch-based methods attain around 53%, but the GAN, 41.4%.[3] Another discrepancy is between semi-supervised and supervised. SEMI-ROTATION-10% and SEMI-EXEMPLAR-10% perform $1 - 1.5\%$ worse than SUP-100% with fine-tuning, but $4 - 5\%$ worse with linear evaluation. The self-supervised methods extract useful representations, just without linear separability. Previous works (Kornblith et al., 2019; Kolesnikov et al., 2019) claim that linear evaluation results are sensitive additional factors that we do not vary, such as ResNet version or pre-training regularization parameters. We conclude that linear evaluation is a poor proxy for overall reduced sample complexity.

## 3.5 ANALYSIS

**Per-Task Metric** In previous works, some datasets are reported using alternative metrics to top-1 accuracy, namely: mean-per-class accuracy (for Caltech101) or Cohen's quadratic kappa (for Retinopathy). We study whether these metrics reveal different results to top-1 accuracy. We find that, although there are some minor ranking differences, the overall picture remains unchanged. Kendall's ranking correlation scores, with respect to top-1 accuracy, are: 1.0 (mean per-class accuracy), 0.97 (quadratic kappa). Appendix D shows the full rankings according to each metric.

**Metric Aggregation Across Tasks** Mean top-1 accuracy across tasks creates an implicit weighting. First, some tasks use the same input data (e.g Clevr-Count and Clevr-Dist), thus upweighting those domains. Second, the task groups differ in size. Third, the tasks exhibit different performance ranges across methods. Therefore, we compare seven different ranking aggregation strategies: (i) Macro-average across datasets (grouping tasks with the same input data). (ii) Macro-average across groups (merging tasks in the same group, ● NATURAL, etc.). (iii) Geometric mean. (iv) Average rank. (v) Average rank after small perturbation of the scores with noise. (vi) Robust (binned) average rank. (vii) Elimination rank – equivalent to an "Exhaustive Ballot".

All strategies have a high Kendall's correlation with the vanilla mean across tasks is high (above 0.9). The most dissimilar strategy on the 1000-example datasets is the perturbed average rank, with correlation 0.91. Therefore, we use mean top-1 accuracy because it is simple, interpretable, and can be computed for each method independently, unlike rank-based aggregation. See Appendix E for details.

---

[3]Surprisingly, fine-tuning the conditional GAN is slightly worse than linear evaluation. Fine-tuning also involves training a linear head from-scratch. It seems that while doing so, the representations in the rest of the model are degraded slightly.

**Representative Subset** For rapid-prototyping, a representative subset of VTAB may be useful. We compute the rank correlation between the mean scores produced by each $\binom{20}{5}$ subsets of five tasks, and the full suite. Appendix F contains the results. The top-5 subsets tend to span different domains, but differ to each other. Although using a subset might be useful for screening models, they were computed using our particular set of models, and may correlate less well in other experiments. Using subsets also increases the risk of meta-overfitting, which VTAB aims to avoid by having many tasks.

**Limiting Evaluation Cost** The cost is near linear in the schedule length. To determine a minimal, yet meaningful, schedule, we sweep over schedules ranging from $40$ to $40,000$ steps, with batch size $512$ on all datasets. Fig. 6 summarizes the results, details Figs. 19 and 20, Appendix M. Most runs reach their optimal performance within $1,000$ steps (updates), and do not improve significantly when trained for longer.

We fine-tune the models using 16 Google Cloud TPU-v3 hardware accelerators. We conduct an additional experiment to assess whether our results can be reproduced with a more basic hardware setup. We evaluate SUP-100% on all VTAB tasks ($1,000$ examples) using a single NVidia P100 GPU with $1,000$ steps, 64 images per mini-batch, and a learning rate of $0.01$ . In this experiment we match the mean performance of the same model reported in Table 1.

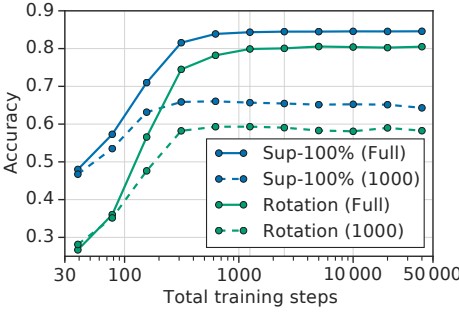

## 4 RELATED WORK

Figure 6: Performance of two methods under various evaluation time constraints.

**Vision Benchmarks** The Visual Decathlon (Rebuffi et al., 2017) contains ten classification tasks: Omniglot, and nine using natural images. Four overlap with VTAB, but VTAB includes several other domains and tasks. Importantly, these benchmarks have opposite evaluation protocols: The Visual Decathlon allows direct joint optimization on the tasks, but forbids external data. By contrast, VTAB forbids multi-task learning, but permits transfer from any arbitrary dataset — this simulates one-off training of representations which are then applied to novel tasks. Further, VTAB considers a low-sample regime (1000-examples), which reflects performance under a reasonable labelling budget. We conduct experiments showing that the methods ranked according to VTAB are more likely to transfer to new tasks, than those ranked according to the Visual Decathlon (Appendix N).

The Facebook AI SSL challenge (Goyal et al., 2019b) was proposed to evaluate self-supervised models. It contains two datasets of natural images, with four tasks, including classification, detection, and low-label classification. The original paper (Goyal et al., 2019a) also includes navigation and surface normal estimation. In contrast, VTAB uses classification tasks only to admit task-independent implementations. VTAB attains diversity with many tasks from alternative domains and with different semantics (localization, counting, etc.). VTAB experiments include self-supervised methods, but are not restricted to them. Indeed, a challenge posed by the VTAB benchmark is to outperform supervised ImageNet models. Most importantly, Facebook SSL requires transfer via a shallow network stacked on a fixed CNN, whereas VTAB permits any transfer, focussing only on performance with few samples.

Meta-Dataset (Triantafillou et al., 2019) contains natural image classification tasks. The protocol differs to VTAB: most of Meta-Dataset's evaluation datasets are also in the training set. The train/test split is instead made across classes. Meta-Dataset is designed for few-shot learning, algorithms that use considerably fewer than 1000 samples. This regime requires different solutions to VTAB.

**Other ML Benchmarks** Benchmark suites have driven progress in other areas of machine learning, such as the GLUE (Wang et al., 2018) and the NL Decathlon (McCann et al., 2018) in NLP, or the Atari suite (Mnih et al., 2013) and Mujoco environments (Todorov et al., 2012) in RL. In NLP, the GLUE Benchmark (Wang et al., 2018) has been a central benchmark used to measure progress of BERT (Devlin et al., 2018) and variants.

**Representation Learning Evaluation** Popular evaluations for representation learning are linear/MLP and semi-supervised. In linear/MLP evaluation, widely used in for self-supervised representations (Doersch et al., 2015; Zhang et al., 2016; Noroozi & Favaro, 2016; Doersch & Zisserman,

2017), the weights of a pre-trained network are frozen, and a linear layer/MLP is trained on top to predict the labels. Evaluation is often performed on the same dataset that was used to train the representations, but using all labels, defeating the goal of sample efficiency. A semi-supervised protocol is more realistic and performs better (Zhai et al., 2019). This evaluation is performed on a single dataset, by contrast VTAB concerns transfer to *unseen* tasks. Linear evaluation is sensitive to small upstream training details, such as the ResNet version or regularization, that make little difference when training end-to-end (Kornblith et al., 2019; Kolesnikov et al., 2019). Other intrinsic scores have been proposed to measure the disentanglement of representations (Locatello et al., 2019), or the mutual information between the input and representations (Hjelm et al., 2018). Both were shown to be weakly correlated with representation utility (Locatello et al., 2019; Tschannen et al., 2019).

**Generative Models Evaluation**  Generative models are usually evaluated by generative quality, a difficult goal with numerous proxy metrics, such as reconstruction error, Frechet Inception Distance (Heusel et al., 2017), Inception score (Salimans et al., 2016), precision-recall (Sajjadi et al., 2018). log likelihood (for auto-regressive models), or the evidence lower bound (for VAEs). Sometimes generative models are also evaluated using linear classification (Radford et al., 2016; Donahue et al., 2017; Dumoulin et al., 2017), or semi-supervised learning (Kingma et al., 2014; Narayanaswamy et al., 2017; Tschannen et al., 2018).

## 5 DISCUSSION

Our results illuminate the current state of representation learning. First, ImageNet labels provide a strong baseline, especially for natural datasets. However, on specialized datasets e.g. medical, distant domains, or tasks that require structured understanding, their value is reduced. Research on how to train on more diverse data sources, such as video, may be the key to moving beyond the "ImageNet domain" towards general, more human-like, visual representations.

Second, generative models — at least the common ones — appear not to be competitive representation learners.[4] Generation may be a useful endeavour in of itself, but if generative losses are used as a means towards representing data, then more explicit representation evaluation may assist progress.

Third, self-supervised learning appears promising, being the best-performing unsupervised method. When combined with a few labels ($10\%$), these methods perform almost as well as fully supervised ImageNet pre-training. Self-supervised even provides some value on top of ImageNet labels *on the same data*. We observe modest, yet significant, improvements over pure supervised learning, especially on tasks that require structured understanding. ImageNet labels encode lots of useful information, but also encode invariances that a universal representation should instead be sensitive to. For example, the distance to an object.

Representation learning is inherently tied to transfer learning; without a transfer strategy representations cannot be used for new tasks. Here, we elect fine-tuning since it performs better than the popular linear-model alternative, and is known to be less brittle. However, it would be interesting to research alternative strategies from domain adaptation (Wang & Deng, 2018), residual adapters (Rebuffi et al., 2017), transfusion (Raghu et al., 2019), and other fields.

VTAB may be used to optimize many design choices involved in learning generalizable representations. Here, we study upstream training loss. We therefore control other factors such as hyperparameter sweep size, architecture, transfer algorithm, downstream preprocessing, and upstream training data. Varying these factors, and others, to improve VTAB is valuable future research. However, the effect of each factor should be isolated from other confounders, such as hyperparameter sweep size. The only approach that is out-of-bounds is to condition the algorithm explicitly on the test tasks, since this would compromise representation generalization.

Code and data to run the Visual Task Adaptation Benchmark is made available, and progress is publicly monitored at [anonymous-url]. All pre-trained models presented are also released. We hope that VTAB drives representation learning towards the goal of making deep learning accessible to the many applications without vast labelling budgets.

---

[4]Concurrent work, BigBiGAN (Donahue & Simonyan, 2019) claims better representations, this would be interesting to evaluate in future work.

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

## A   TASKS

| Category | Dataset | Train size | Classes | Reference |
|---|---|---|---|---|
| • Natural | Caltech101 | 3,060 | 102 | Li et al. (2006) |
| • Natural | CIFAR-100 | 60,000 | 100 | Krizhevsky (2009) |
| • Natural | DTD | 3,760 | 47 | Cimpoi et al. (2014) |
| • Natural | Flowers102 | 2,040 | 102 | Nilsback & Zisserman (2008) |
| • Natural | Pets | 3,680 | 37 | Parkhi et al. (2012) |
| • Natural | Sun397 | 87,003 | 397 | Xiao et al. (2010) |
| • Natural | SVHN | 73,257 | 10 | Netzer et al. (2011) |
| • Specialized | EuroSAT | 21,600 | 10 | Helber et al. (2019) |
| • Specialized | Resisc45 | 25,200 | 45 | Cheng et al. (2017) |
| • Specialized | Patch Camelyon | 294,912 | 2 | Veeling et al. (2018) |
| • Specialized | Retinopathy | 46,032 | 5 | Kaggle & EyePacs (2015) |
| • Structured | Clevr/count | 70,000 | 8 | Johnson et al. (2017) |
| • Structured | Clevr/distance | 70,000 | 6 | Johnson et al. (2017) |
| • Structured | dSprites/location | 663,552 | 16 | Matthey et al. (2017) |
| • Structured | dSprites/orientation | 663,552 | 16 | Matthey et al. (2017) |
| • Structured | SmallNORB/azimuth | 36,450 | 18 | LeCun et al. (2004) |
| • Structured | SmallNORB/elevation | 36,450 | 9 | LeCun et al. (2004) |
| • Structured | DMLab | 88,178 | 6 | Beattie et al. (2016) |
| • Structured | KITTI/distance | 5,711 | 4 | Geiger et al. (2013) |

Table 2: Description of the datasets used for the tasks in VTAB.

Table 2 provides statistics and references for all of the tasks in VTAB.

We now provide a brief description of each task.

**Caltech101** (Li et al., 2006) The task consists in classifying pictures of objects (101 classes plus a background clutter class), including animals, airplanes, chairs, or scissors. The image size varies, but it typically ranges from 200-300 pixels per edge.

**CIFAR-100** (Krizhevsky, 2009) The task consists in classifying natural images (100 classes, with 600 training images each). Some examples include apples, bottles, dinosaurs, and bicycles. The image size is 32x32.

**DTD** (Cimpoi et al., 2014) The task consists in classifying images of textural patterns (47 classes, with 120 training images each). Some of the textures are banded, bubbly, meshed, lined, or porous. The image size ranges between 300x300 and 640x640 pixels.

**Flowers102** (Nilsback & Zisserman, 2008) The task consists in classifying images of flowers present in the UK (102 classes, with between 40 and 248 training images per class). Azalea, Californian Poppy, Sunflower, or Petunia are some examples. Each image dimension has at least 500 pixels.

**Pets** (Parkhi et al., 2012) The task consists in classifying pictures of cat and dog breeds (37 classes with around 200 images each), including Persian cat, Chihuahua dog, English Setter dog, or Bengal cat. Images dimensions are typically 200 pixels or larger.

**Sun397** (Xiao et al., 2010) The Sun397 task is a scenery benchmark with 397 classes and, at least, 100 images per class. Classes have a hierarchy structure, and include cathedral, staircase, shelter, river, or archipelago. The images are (colour) 200x200 pixels or larger.

**SVHN** (Netzer et al., 2011) This task consists in classifying images of Google's street-view house numbers (10 classes, with more than 1000 training images each). The image size is 32x32 pixels.

**EuroSAT** (Helber et al., 2019) The task consists in classifying Sentinel-2 satellite images into 10 different types of land use (Residential, Industrial, River, Highway, etc). The spatial resolution corresponds to 10 meters per pixel, and the image size is 64x64 pixels.

**Resisc45** (Cheng et al., 2017) The Remote Sensing Image Scene Classification (RESISC) dataset is a scene classification task from remote sensing images. There are 45 classes, containing 700 images each, including tennis court, ship, island, lake, parking lot, sparse residential, or stadium. The image size is RGB 256x256 pixels.

**Patch Camelyon** (Veeling et al., 2018) The Patch Camelyon dataset contains 327,680 images of histopathologic scans of lymph node sections. The classification task consists in predicting the presence of metastatic tissue in given image (i.e., two classes). All images are 96x96 pixels.

**Retinopathy** (Kaggle & EyePacs, 2015) The Diabetic Retinopathy dataset consists of image-label pairs with high-resolution retina images, and labels that indicate the presence of Diabetic Retinopahy (DR) in a 0-4 scale (No DR, Mild, Moderate, Severe, or Proliferative DR).

**Clevr/count** (Johnson et al., 2017) CLEVR is a visual question and answer dataset designed to evaluate algorithmic visual reasoning. We use just the images from this dataset, and create a synthetic task by setting the label equal to the number of objects in the images.

**Clevr/distance** (Johnson et al., 2017) Another synthetic task we create from CLEVR consists of predicting the depth of the closest object in the image from the camera. The depths are bucketed into size bins.

**dSprites/location** (Matthey et al., 2017) The dSprites dataset was originally designed to asses disentanglement properties of unsupervised learning algorithms. In particular, each image is a 2D shape where six factors are controlled: color, shape, scale, rotation, and (x,y) center coordinates. Images have 64x64 black-and-white pixels. This task consists in predicting the x (horizontal) coordinate of the object. The locations are bucketed into 16 bins.

**dSprites/orientation** (Matthey et al., 2017) We create another task from dSprites consists in predicting the orientation of each object, bucketed into 16 bins.

**SmallNORB/azimuth** (LeCun et al., 2004) The Small NORB dataset contains images of 3D-toys from 50 classes, including animals, human figures, airplanes, trucks, and cars. The image size is 640x480 pixels. In this case, we define labels depending on the azimuth (angle of horizontal deviation), in intervals of 20 degrees (18 classes).

**SmallNORB/elevation** (LeCun et al., 2004) Another synthetic task we create from Small NORB consists in predicting the elevation in the image. There are 9 classes, corresponding to 9 different elevations ranging from 30 to 70 degrees, in intervals of 5 degrees.

**DMLab** (Beattie et al., 2016) The DMLab (DeepMind Lab) suite is a set of control environments focused on 3D navigation and puzzle-solving tasks. We collected a classification dataset by running some episodes using a pre-defined policy that collects objects by moving to them. The objects in the environment are grouped into two classes. We record the frame (agent's view) at each state, and the time (number of steps) until the agent reaches the next object. The label space is the cross-product of this object's group and the time taken to reach the next it (bucketed into three groups). This task amounts to predicting the identity and distance of an object in a 3D environment.

**KITTI-Dist** (Geiger et al., 2013) The KITTI task consists in predicting the (binned) depth to the vehicle (car, van, or truck) in the image. There are 4 bins / classes.

# B   HUMAN EVALUATION

We define $P(T)$ informally as "All tasks that a human can solve using visual input alone". Intuitively, the tasks in Appendix A satisfy this property because the task semantics involve simple visual concepts (for a human), and the images contain recognisable objects – either in a natural or artificial environment. However, we also check empirically that humans can solve the types of tasks using in VTAB *from examples alone*. We therefore evaluate human raters on a representative subset of the tasks used in VTAB: Pets (natural images, fine-grained object labels), DTD (natural, textures), Camelyon (specialized, medical images), EuroSAT (specialized, aerial images), DMLab (structured, distance prediction), and Clevr-count (structured, object counting).

For each human-evaluated dataset, raters are given 20 random examples for each class, taken from the training split. Raters are asked to classify between 50 and 100 images each, for a total of 1K images (except for DMLab: 534 images), randomly taken from the test split, based on the provided training examples. Beside the examples, no hints nor explanations are given on the nature of the tasks. The raters have to deduce which properties of each image should be used to classify those, for example: the breed of the animal, the distance between the camera and the objects, or the number of objects. The raters are asked not to exchange on the tasks, so each rater produces independent work.

All datasets rated by humans are the same as the one rated by the models, except for DMLab, where we only assess the distance prediction aspect and not object type. This is because there are too many object types for the raters to learn the two groups of objects from 20 examples per class. Therefore, we asses distance-prediction alone. The human DMLab task contains only 3 classes, each of which contains many object types, and differ only in object distance.

Note that this evaluation is not meant to quantify the relative performances of humans and machines, since protocol differences e.g. number of training examples, render the performance incomparable. Instead, it is meant as a verification that the kinds of domains (natural images, aerial imagery, etc.) and semantics (object type classification, localization, counting, etc.) are possible to learn from examples alone using human-level visual representations.

|  | Random Guess | Human |
|---|---|---|
| Pets | 2.7% | 63.1% |
| DTD | 2.1% | 64.0% |
| Camelyon | 50% | 68.0% |
| EuroSAT | 10% | 86.5% |
| DMLab | 33.3% | 49.0% |
| Clevr-count | 12.5% | 99.0% |

Table 3: Human evaluation scores, measured using mean-per-class accuracy.

We measure human performance using mean-per-class accuracy. This is because the human training sets are class-balanced, so the raters cannot learn the class priors which algorithms that see an i.i.d sample of the training set could. Table 3 shows the results. The results indicate that some tasks are harder due to more subtle distinctions (Camelyon) or noise (DMLab), and some very easy (Clevr-count). However, in all cases the raters perform significantly better than random guessing. This demonstrates that the types of tasks used in VTAB are ones for which human-like visual representations are useful to solve them, using few labels and visual-input alone.

## C  NETWORK ARCHITECTURES

All of the methods (except for *BigGAN* models) we evaluate in the paper use the standard ResNet50-v2 (He et al., 2016) architecture.

This architecture consists of 50 convolutional layers. The representation produced by this model has 2048 dimensions. Exact details of this architecture can be found in (He et al., 2016).

In the *Cond-BigGAN* and *Uncond-BigGAN* models we use publicly available[5] implementation by Lucic et al. (2019) and (Chen et al., 2019) of the custom ResNet-like architecture proposed and described in (Brock et al., 2019). It has 1536 dimensions in the final representation layer.

We use a specialized procedure for evaluating patch-based models (*Relative Patch Location* (Doersch et al., 2015) and *Jigsaw* (Noroozi & Favaro, 2016)). These models use ResNet50 model with the overall stride reduced from 64 to 16 (by substituting the first and the last strided convolution of the standard ResNet50 model by a convolution with stride one). During the adaptation phase, we apply ResNet50 with reduced stride in the following way (assuming that input image size is $224 \times 224$):

- Perform central crop of size $192 \times 192$.
- Cut image into $3 \times 3$ patches of size $64 \times 64$.
- Apply the ResNet50 model independently to every patch.
- Output the final representation and element-wise average of 9 individual patch representations.

---

[5]`https://github.com/google/compare_gan/blob/master/compare_gan/`
`architectures/resnet_biggan.py`

# D    ALTERNATIVE METRICS

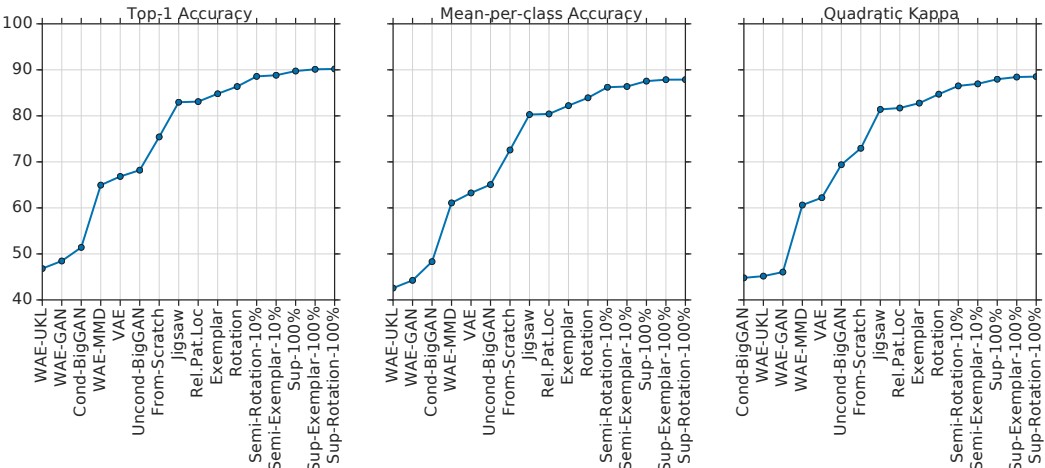

Figure 7: Ranking of the methods using the average scores across datasets (validation split) using three different metrics: top-1 accuracy (left), mean-per-class accuracy (center), Cohen's quadratic kappa (right). The methods on the x-axis are sorted according to the highest scores according to each metric. Although there are some minor changes in the ranking between top-1 and Cohen's quadratic kappa, the overall performance of groups of methods remains unchanged.

Figure 7 shows different test metric scores (top-1 accuracy, mean-per-class accuracy, and Cohen's quadratic kappa), averaged across all datasets in the benchmark, for the different methods studied. The methods were ranked in the x-axis according to their score in each metric. Observe that top-1 and mean-per-class accuracy give exactly the same ranking. There are subtle differences when using Cohen's quadratic kappa, but the overall picture remains unchanged: supervised methods outperform semi-supervised methods by a small margin, followed by self-supervised methods, etc. Kendall's ranking correlation scores, with respect to top-1 accuracy, are: 1.0 for mean per-class accuracy, and 0.97 for Cohen's quadratic kappa. This confirms that our conclusions hold even if we are not using the standard metrics for a few datasets.

# E   ALTERNATIVE WEIGHTING AND RANKING SCHEMES FOR MODELS

Throughout, we use (unweighted) mean top-1 accuracy across all tasks to rank the different models. This assumes that the samples represent our desired task distribution $P_T$ in an unbiased manner. However, there may be other sensible weighting schemes that are not uniform. Here we explore three alternative weighting schemes (i) assigning equal weight to every dataset – there are three data sets that are used in two tasks, and the others are used in one task, (ii) assigning equal weight to each task group: ●NATURAL, ●SPECIALIZED, and ●STRUCTURED, and (iii) the weighting scheme introduced in Balduzzi et al. (2018).

Another issue inherent in (weighted or unweighted) mean accuracy is that the mean accuracies of individual tasks vary significantly depending on the task's difficulty. Since maximum accuracy is bounded, this may limit the range of performances, implicitly upweighting tasks with more "headroom". Therefore, we explore a few simple alternative ranking strategies (see Dwork et al. (2001) for an introduction raking aggregation methods):

   (i) Ranking according to the geometric mean.
  (ii) The average rank obtained by ranking the models for each task according to accuracy and then computing the average rank across tasks.
 (iii) A noise-perturbed version of (i) in which the accuracies are perturbed by Gaussian noise with variance 1.0 prior to ranking.
 (iv) A robust variant of the average rank, where, prior to averaging ranks across tasks, the accuracy is binned into buckets of size $1\%$ and all models in the same bucket obtain the same rank.
  (v) An elimination ranking scheme equivalent to the "Exhaustive Ballot" voting system, see e.g. (Shahandashti, 2016).

We measure the agreement between these ranking strategies using the Kendall rank correlation coefficient (Kendall, 1945). To account for the training/evaluation stochasticity in VTAB, we sample (independently for every pair of model and task) one out of the three test-set repetitions, and compute the rank correlation between ranking according to the mean accuracy and each alternative ranking. We average over 100 such samples. The results are in Table 4. The mean rank correlation between the ranking according to the mean across tasks correlates very well with alternative ranking schemes. In particular, weighted means for 1000 samples and the full dataset as well as the geometric mean have a rank correlation with the ranking according to the mean that exceeds 0.93. The agreement with different types of average rank somewhat lower for 1000 samples, but still considerable. We hence conclude that the mean accuracy across tasks is a fairly robust metric to rank models in VTAB.

| Ranking Method | 1000 Samples | Full Dataset |
|---|---|---|
| Reweighted data sets | $0.979 \pm 0.017$ | $0.977 \pm 0.018$ |
| Reweighted groups | $0.988 \pm 0.013$ | $0.992 \pm 0.011$ |
| Balduzzi et al. (2018) weighting | $0.925 \pm 0.022$ | $0.945 \pm 0.023$ |
| Geometric mean | $0.963 \pm 0.022$ | $0.969 \pm 0.020$ |
| Average rank | $0.922 \pm 0.022$ | $0.951 \pm 0.017$ |
| Average rank (perturbed) | $0.912 \pm 0.028$ | $0.931 \pm 0.031$ |
| Average rank (robust) | $0.926 \pm 0.021$ | $0.971 \pm 0.020$ |
| Elimination rank | $0.913 \pm 0.029$ | $0.935 \pm 0.020$ |

Table 4: Kendall rank correlation coefficient (Kendall, 1945) (with standard deviation) measuring the agreement of the ranking of models according to the mean accuracy across tasks with different types of weighted means, the geometric mean, and different types of average ranks. Agreement of ranking according to mean accuracy with alternative ranking schemes is high.

## F  REPRESENTATIVE SUBSET OF TASKS

We explore another direction related to ranking of models: Which subset of tasks is most representative of the performance of the entire benchmark? Such a subset allows for cheap iteration which is beneficial during early development of new models.

We search the most representative 5 tasks from the full set of 20 tasks by performing exhaustive search over all $\binom{20}{5}$ subsets. For each subset we compute the mean accuracy and compute the Kendall rank correlation coefficient between the resulting ranking of models and the ranking according to the mean over all tasks (averaged over 10 trials sampled as described in Appendix E). Table 5 shows the subsets which produce the highest rank correlation. There are many subsets which lead to an excellent agreement with the ranking according to the full mean. These subsets can be quite different provided that they are sufficiently diverse.

To assess how well we can expect this approach generalize to unseen models, we perform the following experiment. For each pair of models, we perform subset selection based on the accuracies of the remaining models, and check whether the ranking of the left out pair based on the mean across all data sets agrees with the ranking of the pair according to the mean over the subset. The mean outcome of this binary test across all pairs is an estimate of the probability for correctly ranking an unseen pair of models: $0.940$ for the full data sets and $0.972$ for 1000 examples. The subset selection hence generalizes to an unseen pair of models with high probability.

The representative subsets in Table 5 may be used for rapid prototyping of new methods before running the full VTAB. However, a caveat to the above analyses is that we evaluate the ranking of a very diverse set of models, from those whose performance is worse than from-scratch training to performance combinations of self-supervision and supervision. Therefore, the ranking is reasonably robust. To make fine-grained distinctions between models of a similar class, the representative subsets above may be less reliable. Repeated iteration on just a few tasks exposes one more to the risk of meta-overfitting.

| Dataset | Rank Correlation | Subset |
|---------|------------------|--------|
| 1000 | $0.983 \pm 0.017$ | Caltech101, EuroSAT, Pets, Camelyon, Resisc45 |
|  | $0.980 \pm 0.012$ | DTD, EuroSAT, Flowers102, Pets, SVHN |
|  | $0.978 \pm 0.013$ | Caltech101, DMLab, EuroSAT, Flowers102, Camelyon |
|  | $0.978 \pm 0.011$ | Caltech101, EuroSAT, Flowers102, Pets, Camelyon |
|  | $0.975 \pm 0.013$ | Cifar100, Clevr-Count, dSpr-Orient, Flowers102, Pets |
| full | $0.987 \pm 0.012$ | Clevr-Dist, DTD, Camelyon, Sun397, SVHN |
|  | $0.987 \pm 0.012$ | Cifar100, DMLab, dSpr-Loc, Flowers102, SVHN |
|  | $0.983 \pm 0.020$ | Clevr-Dist, Clevr-Count, Retinopathy, DTD, Flowers102 |
|  | $0.983 \pm 0.017$ | Retinopathy, dSpr-Loc, Flowers102, sNORB-Azim, SVHN |
|  | $0.982 \pm 0.016$ | Clevr-Dist, DTD, EuroSAT, Camelyon, Sun397 |

Table 5: Task subsets of size 5 that lead to the largest rank correlation with the mean accuracy across all tasks. There are many different subsets that lead to an high rank correlation with the full mean.

# G  LIGHTWEIGHT EXPERIMENTS

| | | Caltech101 | CIFAR-100 | DTD | Flowers102 | Pets | SVHN | Sun397 | Camelyon | EuroSAT | Resisc45 | Retinopathy | Clevr-Count | Clevr-Dist | DM-Lab | KITTI-Dist | dSpr-Loc | dSpr-Ori | sNORB-Azim | sNORB-Elev | Mean |
|---|---|---|---|---|---|---|---|---|---|---|---|---|---|---|---|---|---|---|---|---|---|
| **1000** | WAE-GAN | 29.9 | 7.8 | 6.5 | 12.0 | 7.5 | 42.4 | 3.2 | 69.0 | 59.1 | 14.4 | 73.4 | 32.4 | 55.1 | 25.0 | 54.7 | 70.1 | 17.8 | 27.0 | 25.4 | 33.3 |
| | WAE-UKL | 29.3 | 8.2 | 7.2 | 13.1 | 7.8 | 36.6 | 3.3 | 69.8 | 61.2 | 16.4 | 73.6 | 32.1 | 55.2 | 22.4 | 50.9 | 72.9 | 23.0 | 27.5 | 28.1 | 33.6 |
| | VAE | 34.3 | 9.9 | 10.3 | 18.2 | 9.4 | 49.3 | 4.5 | 72.6 | 73.1 | 23.5 | 67.7 | 36.8 | 61.2 | 26.7 | 42.9 | 89.9 | 23.0 | 35.6 | 37.5 | 38.2 |
| | WAE-MMD | 35.5 | 10.7 | 8.0 | 18.4 | 9.3 | 58.7 | 4.9 | 73.8 | 73.1 | 22.4 | 73.2 | 34.4 | 58.1 | 26.3 | 55.4 | 86.9 | 26.3 | 31.6 | 28.1 | 38.7 |
| | Cond-BigGAN | 62.1 | 21.1 | 38.5 | 60.0 | 23.4 | 59.4 | 11.9 | 77.2 | 30.6 | 48.2 | 73.6 | 12.3 | 24.5 | 28.1 | 41.0 | 54.5 | 54.7 | 33.2 | 32.5 | 41.4 |
| | From-Scratch | 38.2 | 13.2 | 20.6 | 40.2 | 12.1 | 66.1 | 5.0 | 75.6 | 85.0 | 42.6 | 72.6 | 38.8 | 55.2 | 26.6 | 42.7 | 78.4 | 39.2 | 30.1 | 37.6 | 43.1 |
| | Un.C.-BigGAN | 58.7 | 15.2 | 34.1 | 49.9 | 18.7 | 65.4 | 9.4 | 76.3 | 74.2 | 35.7 | 66.1 | 44.7 | 55.8 | 28.7 | 37.2 | 76.8 | 48.4 | 33.9 | 40.3 | 45.8 |
| | Jigsaw | 66.7 | 18.9 | 51.4 | 66.1 | 37.5 | 55.1 | 12.1 | 76.0 | 91.5 | 66.2 | 72.4 | 42.8 | 55.9 | 30.5 | 68.2 | 69.5 | 35.0 | 44.9 | 36.3 | 52.5 |
| | Rel.Pat.Loc. | 68.5 | 19.1 | 52.2 | 69.0 | 41.3 | 60.9 | 11.1 | 77.5 | 92.6 | 65.4 | 70.7 | 43.5 | 59.6 | 33.6 | 68.2 | 70.7 | 29.3 | 47.2 | 35.2 | 53.5 |
| | Exemplar | 69.6 | 11.0 | 48.9 | 69.9 | 42.3 | 87.1 | 13.7 | 79.1 | 94.7 | 67.3 | 72.5 | 54.8 | 60.9 | 36.7 | 75.5 | 91.8 | 44.3 | 42.1 | 40.3 | 58.0 |
| | Rotation | 77.6 | 27.3 | 51.4 | 68.8 | 48.3 | 89.4 | 12.9 | 80.4 | 93.5 | 69.4 | 70.9 | 45.6 | 60.9 | 40.4 | 75.9 | 89.6 | 49.5 | 52.2 | 44.3 | 60.4 |
| | Semi-Rot-10% | 88.1 | 48.1 | 64.3 | 87.6 | 84.6 | 87.2 | 27.0 | 81.4 | 95.6 | 79.5 | 73.0 | 40.9 | 55.3 | 37.9 | 57.4 | 87.5 | 53.5 | 51.9 | 35.7 | 65.1 |
| | Semi-Ex-10% | 88.6 | 53.2 | 60.8 | 86.8 | 85.3 | 88.0 | 29.0 | 83.2 | 95.2 | 77.3 | 71.7 | 42.3 | 57.4 | 36.7 | 71.4 | 74.9 | 53.9 | 52.7 | 32.3 | 65.3 |
| | Sup-100% | 91.0 | 57.0 | 66.0 | 88.6 | 89.9 | 87.3 | 34.4 | 80.6 | 95.3 | 80.8 | 73.2 | 41.0 | 56.1 | 36.3 | 70.6 | 85.7 | 46.0 | 45.7 | 35.4 | 66.4 |
| | Sup-Ex-100% | 90.2 | 56.7 | 67.3 | 88.3 | 89.7 | 88.9 | 34.5 | 83.7 | 95.7 | 80.5 | 72.4 | 37.8 | 56.0 | 37.8 | 77.7 | 84.3 | 53.9 | 49.6 | 40.5 | 67.7 |
| | Sup-Rot-100% | 91.7 | 53.7 | 69.5 | 90.8 | 88.1 | 88.5 | 32.8 | 83.4 | 96.0 | 82.0 | 71.1 | 47.3 | 57.2 | 36.6 | 77.1 | 88.3 | 52.1 | 51.6 | 33.7 | 68.0 |
| **Full** | WAE-UKL | 41.7 | 23.2 | 12.3 | 17.2 | 12.3 | 65.5 | 12.0 | 76.4 | 78.1 | 36.8 | 73.6 | 44.5 | 67.8 | 36.7 | 55.1 | 98.1 | 51.4 | 35.9 | 51.0 | 46.8 |
| | WAE-GAN | 42.0 | 24.8 | 8.7 | 15.5 | 13.1 | 78.2 | 12.8 | 77.1 | 81.5 | 38.4 | 73.6 | 52.2 | 70.2 | 37.3 | 62.3 | 97.7 | 49.9 | 33.4 | 52.2 | 48.5 |
| | Cond-BigGAN | 0.1 | 56.3 | 44.8 | 68.8 | 31.6 | 91.4 | 44.9 | 81.3 | 94.5 | 76.5 | 75.3 | 12.4 | 24.5 | 51.4 | 49.7 | 6.2 | 7.4 | 80.6 | 79.2 | 51.4 |
| | WAE-MMD | 50.8 | 38.8 | 11.0 | 20.8 | 16.2 | 90.9 | 31.6 | 80.6 | 94.1 | 64.8 | 73.8 | 98.1 | 89.3 | 52.6 | 61.6 | 100.0 | 90.2 | 96.3 | 72.4 | 64.9 |
| | VAE | 48.4 | 44.2 | 16.0 | 18.4 | 14.0 | 93.1 | 29.3 | 81.3 | 92.5 | 65.0 | 74.2 | 98.4 | 90.1 | 59.7 | 57.0 | 100.0 | 94.7 | 97.9 | 95.6 | 66.8 |
| | Un.C.-BigGAN | 73.6 | 58.1 | 44.9 | 63.5 | 30.9 | 93.0 | 46.9 | 82.2 | 89.8 | 75.4 | 75.9 | 47.6 | 54.9 | 54.8 | 57.4 | 86.1 | 95.9 | 88.1 | 76.6 | 68.2 |
| | From-Scratch | 55.9 | 64.4 | 31.3 | 50.6 | 23.8 | 96.3 | 52.7 | 81.2 | 96.2 | 86.8 | 76.8 | 99.7 | 89.4 | 71.5 | 68.4 | 100.0 | 96.3 | 99.9 | 91.7 | 75.4 |
| | Jigsaw | 79.1 | 65.3 | 63.9 | 77.9 | 65.4 | 93.9 | 59.2 | 83.0 | 97.9 | 92.0 | 80.1 | 99.6 | 88.6 | 72.0 | 74.7 | 100.0 | 90.3 | 99.9 | 93.6 | 83.0 |
| | R.P.L | 79.9 | 65.7 | 65.2 | 78.8 | 66.8 | 93.7 | 58.0 | 85.3 | 97.8 | 91.5 | 79.8 | 99.5 | 87.7 | 71.5 | 75.0 | 100.0 | 90.4 | 99.7 | 92.6 | 83.1 |
| | Exemplar | 81.9 | 70.7 | 61.1 | 79.3 | 67.8 | 96.7 | 58.2 | 84.7 | 98.5 | 93.5 | 79.0 | 99.8 | 93.3 | 74.7 | 78.2 | 100.0 | 96.5 | 99.9 | 97.4 | 84.8 |
| | Rotation | 88.3 | 73.6 | 63.3 | 83.4 | 71.8 | 96.9 | 60.5 | 86.4 | 98.3 | 93.4 | 78.6 | 99.8 | 93.3 | 76.8 | 82.6 | 100.0 | 96.5 | 99.9 | 98.0 | 86.4 |
| | Semi-Rot-10% | 88.1 | 82.4 | 72.4 | 93.2 | 87.9 | 96.9 | 66.7 | 78.6 | 98.7 | 94.9 | 79.0 | 99.8 | 93.2 | 76.1 | 81.0 | 100.0 | 96.5 | 99.9 | 97.5 | 88.6 |
| | Semi-Ex-10% | 85.3 | 82.7 | 70.5 | 92.2 | 89.0 | 97.0 | 67.4 | 86.0 | 98.6 | 94.7 | 78.8 | 99.8 | 93.1 | 76.8 | 81.5 | 100.0 | 96.5 | 100.0 | 97.8 | 88.8 |
| | Sup-100% | 94.1 | 83.8 | 74.0 | 93.2 | 91.9 | 97.0 | 70.7 | 83.9 | 98.8 | 95.3 | 79.3 | 99.8 | 92.1 | 76.4 | 80.7 | 100.0 | 96.4 | 99.8 | 97.8 | 89.7 |
| | Sup-Ex-100% | 94.4 | 84.1 | 74.5 | 93.4 | 91.8 | 97.1 | 69.4 | 86.7 | 98.6 | 95.1 | 79.5 | 99.8 | 92.7 | 76.8 | 84.0 | 100.0 | 96.4 | 99.8 | 98.0 | 90.1 |
| | Sup-Rot-100% | 94.6 | 84.8 | 75.9 | 94.7 | 91.5 | 97.0 | 70.2 | 85.9 | 98.8 | 94.9 | 79.5 | 99.8 | 92.5 | 76.5 | 82.3 | 100.0 | 96.5 | 100.0 | 98.4 | 90.2 |

Table 6: Top-1 accuracy of all the models evaluated on VTAB in lightweight mode. Each entry represents the median score of three runs with different random seeds evaluated on the test set. Within each dataset-size group (1000-example and full), the methods are sorted from best to worst according to their mean accuracy.

## H  HEAVYWEIGHT EXPERIMENTS

Table 7 shows the full heavyweight results. Table 8 shows recently reported results for each dataset in the literature (where available). The literature results may use substantially more complex architectures, or additional task-specific logic. For example, the Retinopathy result uses three neural networks combined with decision trees, and uses additional information, such as combining the images from two eyes of the same patient.

| | | Caltech101 | CIFAR-100 | DTD | Flowers102 | Pets | SVHN | Sun397 | Camelyon | EuroSAT | Resisc45 | Retinopathy | Clevr-Count | Clevr-Dist | DMLab | KITTI-Dist | dSpr-Loc | dSpr-Ori | sNORB-Azim | sNORB-Elev | Mean |
|---|---|---|---|---|---|---|---|---|---|---|---|---|---|---|---|---|---|---|---|---|---|
| **1000** | From-Scratch | 60.2 | 24.5 | 51.1 | 73.4 | 43.1 | 88.3 | 14.9 | 82.1 | 93.3 | 63.9 | 75.4 | 53.0 | 57.3 | 36.6 | 55.7 | 88.6 | 67.6 | 52.2 | **81.6** | 61.2 |
| | Sup-100% | 89.7 | **52.7** | 65.4 | 94.2 | 90.1 | 91.0 | **35.5** | **87.3** | 96.0 | 84.0 | 78.0 | 59.0 | **63.0** | 44.4 | **80.1** | **89.3** | 70.7 | 57.5 | 54.7 | 72.8 |
| | Sup-Rot-100% | 88.0 | 48.9 | **68.1** | **94.7** | 89.0 | **91.8** | 33.7 | 86.2 | **96.6** | **85.2** | **78.5** | 66.5 | 61.5 | **45.5** | 77.2 | 88.8 | **71.8** | 59.4 | 52.0 | 72.8 |
| | Sup-Ex-100% | **90.4** | 50.4 | 66.1 | 94.3 | **90.6** | 91.3 | 35.2 | 86.7 | 95.8 | 83.3 | 78.4 | **73.8** | 62.4 | 44.1 | 79.7 | 87.5 | 71.1 | **62.8** | 54.8 | **73.6** |
| **Full** | From-Scratch | 74.5 | 77.8 | 67.1 | 85.8 | 70.9 | 97.0 | 70.1 | **91.2** | 98.8 | 94.3 | 82.8 | 99.8 | 96.7 | 76.6 | 68.4 | **100.0** | 96.7 | 99.9 | 94.0 | 86.4 |
| | Sup-100% | 93.0 | 83.4 | 73.7 | 97.3 | 92.7 | 97.5 | 75.6 | 87.3 | 98.8 | 96.1 | 83.4 | 100.0 | **97.0** | 78.8 | 81.0 | **100.0** | **96.8** | 100.0 | 98.5 | 91.1 |
| | Sup-Rot-100% | 93.5 | **84.0** | **76.8** | 97.4 | 92.6 | 97.4 | **75.6** | 86.5 | **99.1** | **96.3** | **83.7** | 100.0 | 96.8 | **79.6** | 82.4 | 100.0 | **96.8** | 100.0 | 96.7 | 91.3 |
| | Sup-Ex-100% | **93.8** | 83.1 | 76.5 | **97.8** | **92.9** | 97.5 | 75.3 | 86.5 | 99.0 | 96.3 | 83.7 | 99.9 | 96.8 | 79.3 | **82.8** | 100.0 | 96.7 | **100.0** | **99.1** | **91.4** |

Table 7: Top-1 accuracy of all the models evaluated on VTAB in heavyweight mode.

| | SUP-EX.-100% | Result | Reference |
|---|---|---|---|
| ● Caltech101 | 90.4 / 95.1[*] | 86.9 / 95.1[*] | Cubuk et al. (2019) / Kornblith et al. (2019) |
| ● CIFAR-100 | 83.1 | 91.7 | Tan & Le (2019) |
| ● DTD | 76.5 | 78.1 | Kornblith et al. (2019) |
| ● Flowers102 | 97.8 | 98.8 | Tan & Le (2019) |
| ● Pets | 92.9 | 95.9 | Huang et al. (2018) |
| ● SVHN | 97.5 | 99.0 | Cubuk et al. (2019) |
| ● Sun397 | 75.3 | 72.0 | Wang et al. (2017) |
| ● Camelyon | 86.5 | 90.6 | Teh & Taylor (2019) |
| ● EuroSAT | 99.0 | 96.4 | Helber et al. (2019) |
| ● Resisc45 | 96.3 | 90.4 | Cheng et al. (2017) |
| ● Retinopathy | 74.7[†] | 85.4[†] | Wang et al. (2017) |

[*] Mean per-class accuracy    [†] Quadratic Kappa

Table 8: Comparison of the best method on the full datasets using heavyweight sweep (SUP-EXEMPLAR-100%) to results published in the literature (where available). For some datasets prior work does not use top-1 accuracy, therefore, we present our the performance of SUP-EXEMPLAR-100% using the same metric.

## I HYPERPARAMETER SWEEPS

**Lightweight sweep** The lightweight mode performs a restricted hyperparameter search for all tasks, permitting fair comparison with few resources. It uses fixed values for most hyperparameters. We set the batch size to 512 and use SGD with momentum of 0.9. When fine-tuning, we do not use weight decay, and when training from scratch we set it to 0.001. We resize all images to $224 \times 224$, except for generative models, where we resize to $128 \times 128$ since training at a higher resolution is currently very challenging.

Lightweight mode sweeps four hyperparameters:

- Learning rate: $\{0.1, 0.01\}$
- Traning schedule: In all cases we decay the learning rate by a factor of 10 after $\frac{1}{3}$ and $\frac{2}{3}$ of the training time, and one more time shortly before the end. We train for $\{2\,500, 10\,000\}$ training steps (*i.e.* model updates).

Note that when training on 1000 examples, we perform model selection using the regular validation set for each task. This setting is somewhat unrealistic, since in practice one would train on the larger validation set. However, we use this setting, which is common in prior art (Oliver et al., 2018), because it significantly reduces evaluation noise. Further, Zhai et al. (2019) show that using a small validation yields the same conclusions.

**Heavyweight sweep** We define a relatively large search-space over relevant hyperparameters for the fine-tuning adaptation phase and perform an individual random search for each downstream task's validation set in that space. The random search consists of 100 independent trials. Specifically, we search:

- Batch-size: $\{128, 256, 512\}$
- Training schedule: In all cases we decay the learning rate by a factor of 10 after $\frac{1}{3}$ and $\frac{2}{3}$ of the training time, and one more time shortly before the end. We train for either
  - any of $\{100, 200, 300, 400, 500\}$ epochs over the dataset, or
  - any of $\{2\,500, 5\,000, 10\,000, 20\,000, 40\,000\}$ training steps (*i.e.* model updates).
- Preprocessing: during training, optionally include random horizontal flipping.
- Preprocessing: during training, optionally include random color distortions.
- Preprocessing: during training, use any of the following techniques for resizing:
  - `res.`: resize the image to 224;
  - `res.|crop`: resize the image to 256 and take a random crop of size 224;
  - `res.sma|crop`: resize the image keeping its aspect ratio such that the smaller side is 256, then take a random crop of size 224;
  - `inc.crop`: "inception crop" from Szegedy et al. (2015);
  - `cif.crop`: resize the image to 224, zero-pad it by 28 on each side, then take a random crop of size 224.
- Preprocessing: for evaluation, either
  - resize the image to one of $\{224, 320, 384\}$, or
  - resize it such that the smaller side is of size $\{256, 352, 448\}$ and then take a central crop of size $\{224, 320, 384\}$.
- Weight decay: we sample log-uniform randomly in $[10^{-5}, 10^{-1}]$.
- Optimizer: we randomly choose between
  - SGD with learning-rate chosen log-uniform randomly in $[0.001, 0.0]$, or
  - Adam with learning-rate chosen log-uniform randomly in $[10^{-5}, 10^{-2}]$.

We then choose the set of hyperparameters with the best performance at the end of training for each task individually.

For both training from scratch, and from a pre-trained supervised ImageNet training, we present the best set of hyperparameters for each task in Figs. 8 to 10 as a red dot. The green background

shows the distribution of hyperparameters which performed within 2% of the best one for each task, providing a sense for the importance of tuning that hyperparameter.

In Figs. 12 and 13, we show the achieved performance for the full distribution of hyperparameters as violin plots for the CIFAR-100 and DMLab downstream tasks, both for the small and full variants. The supervised pre-training not only achieves better accuracy than training from scratch, but it also displays much smaller variance across hyperparameter values.

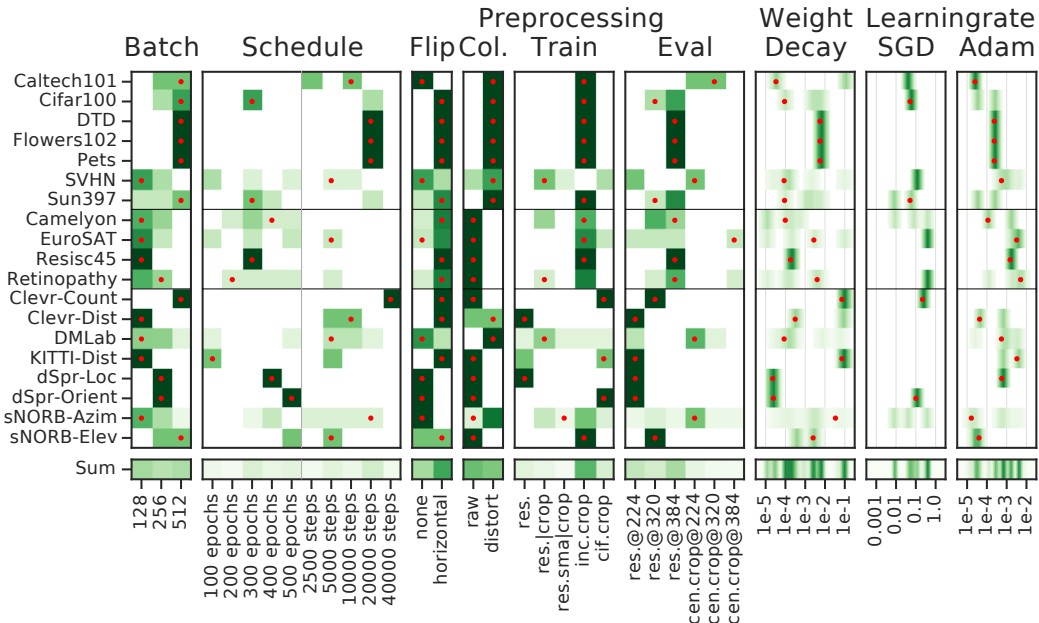

Figure 8: Training from scratch with only 1000 examples. Best hyperparameter values are marked in red and those within 2% in green to show sensitivity.

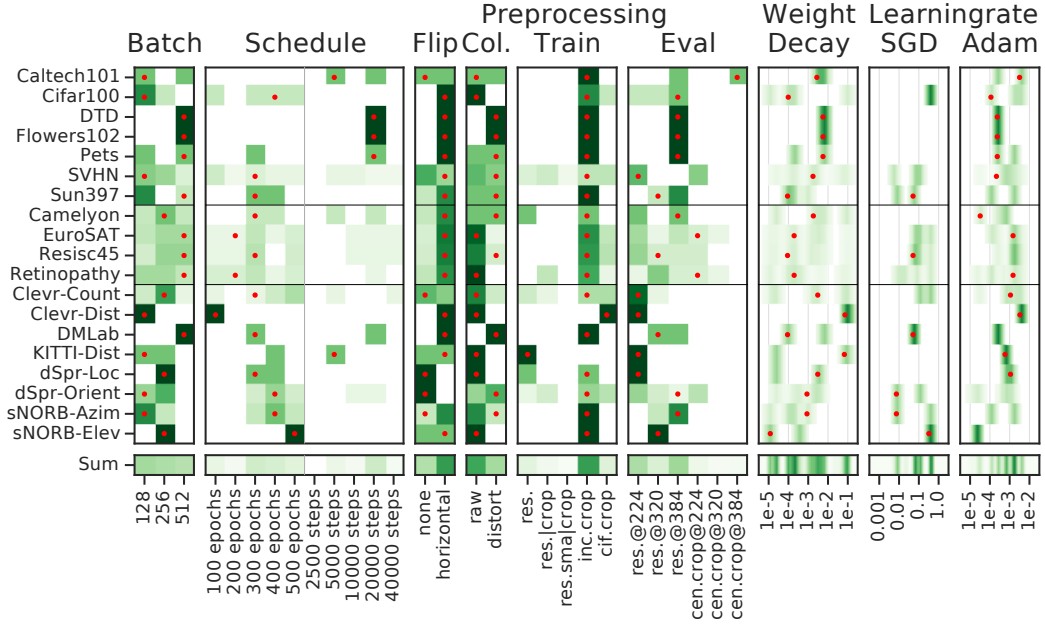

Figure 9: Training from scratch on the full datasets. Best hyperparameter values are marked in red and those within 2% in green to show sensitivity.

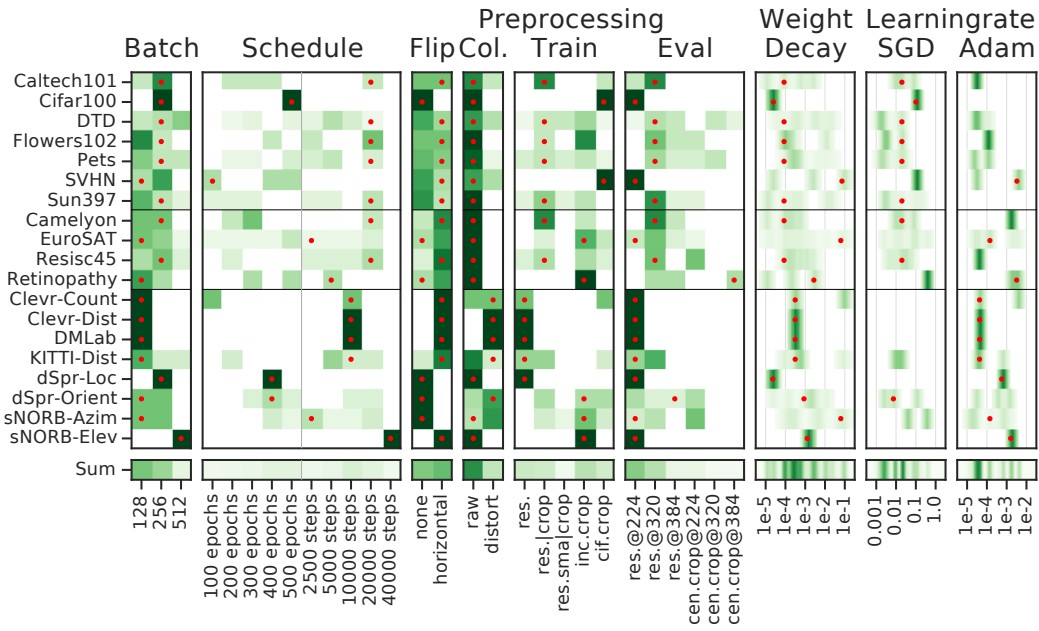

Figure 10: ImageNet supervised representation fine-tuned on only 1000 examples. Best hyperparameter values are marked in red and those within 2% in green to show sensitivity.

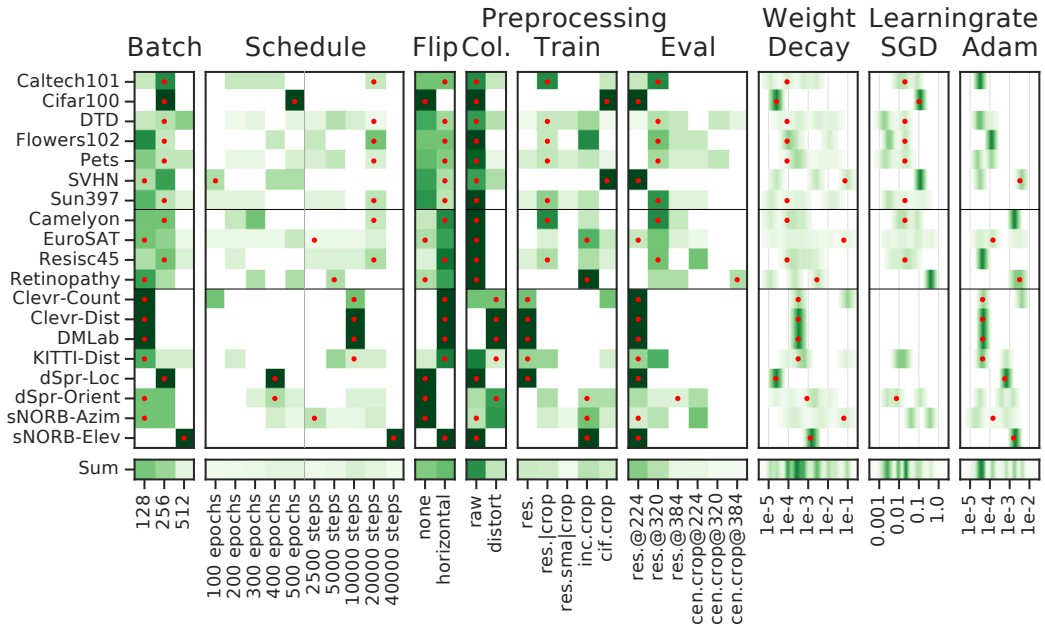

Figure 11: ImageNet supervised representation fine-tuned on the full datasets. Best hyperparameter values are marked in red and those within 2% in green to show sensitivity.

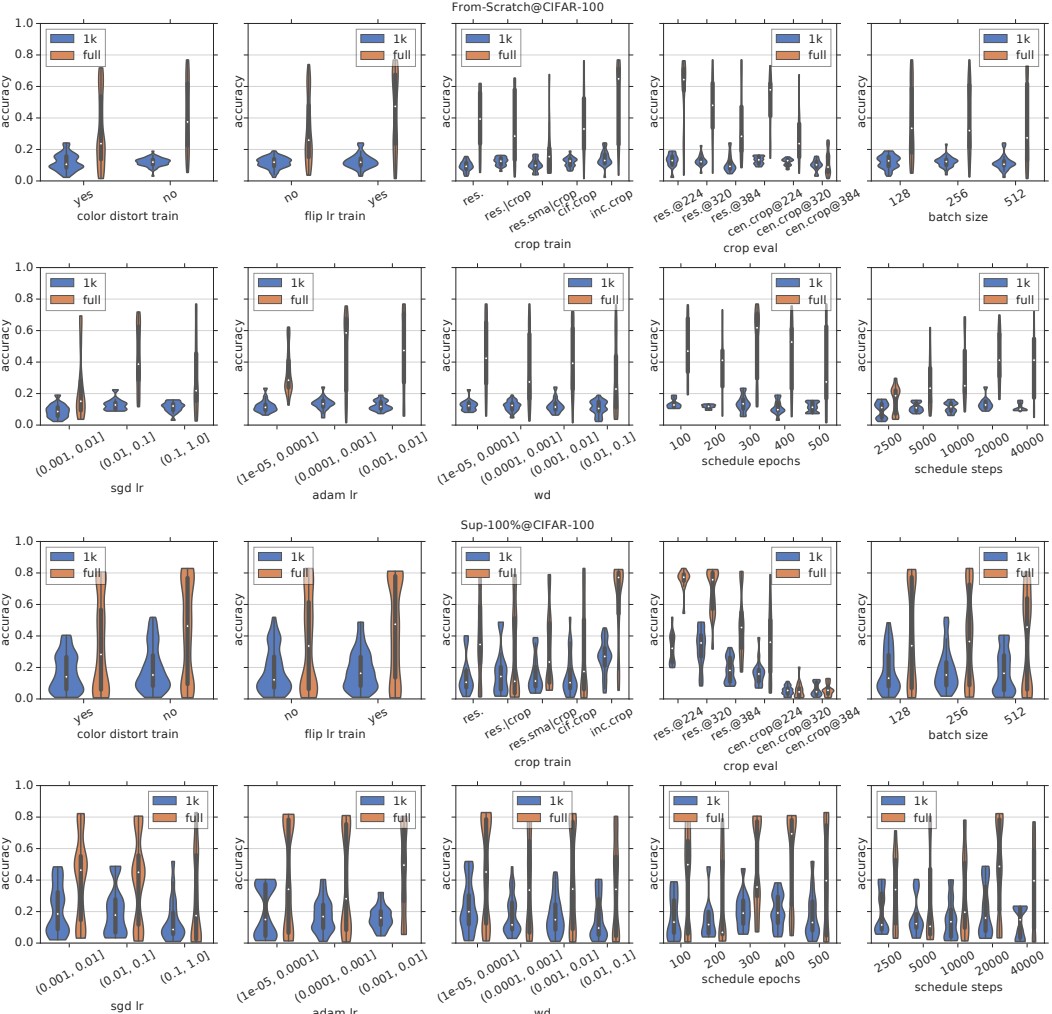

Figure 12: Heavy parameter sweep on CIFAR-100 dataset. Top 2 rows show the results of from scratch algorithm and bottom 2 rows show the results of supervised 100% algorithm. Each plot shows the violin plot of the results with respect to a given parameter. Here we show the results evaluated on full datasets and 1k datasets.

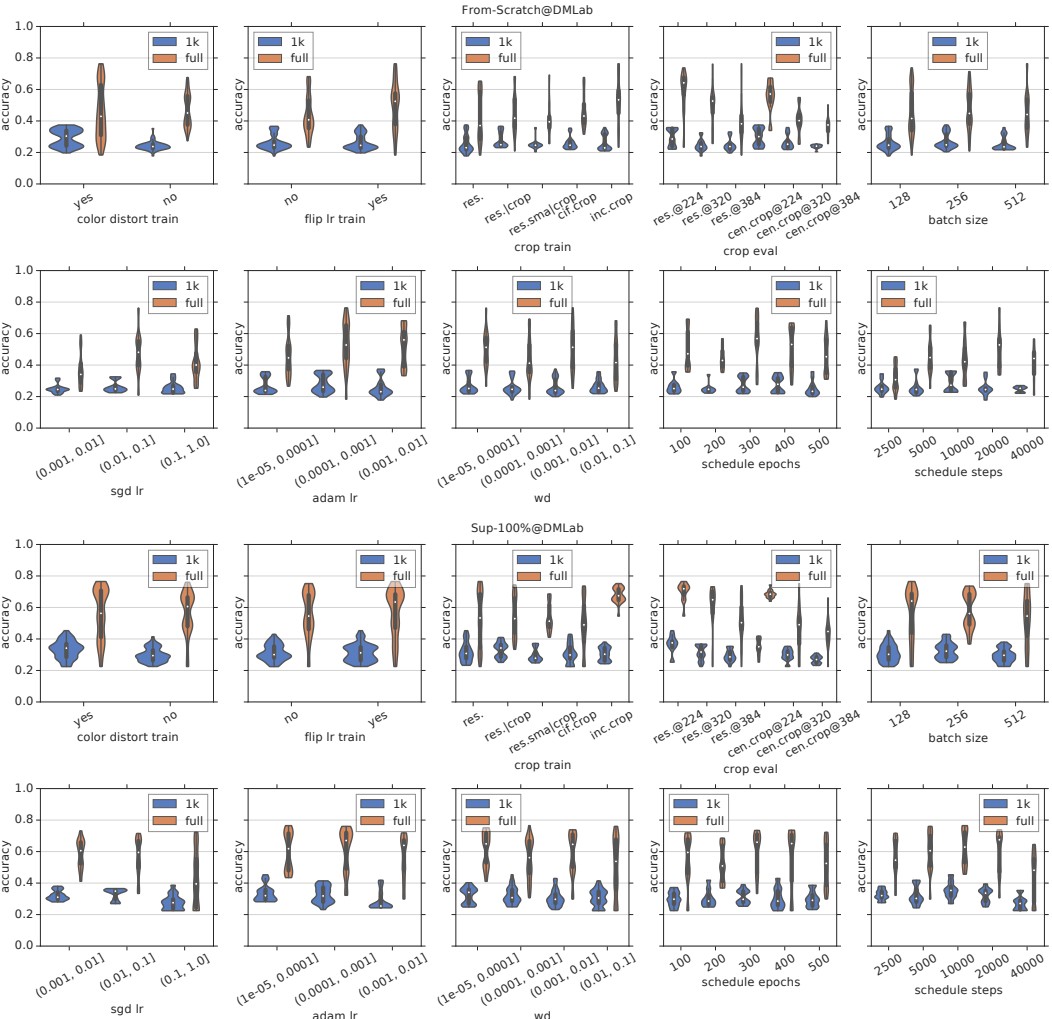

Figure 13: Heavy parameter sweep on DMLab dataset. Top 2 rows show the results of from scratch algorithm and bottom 2 rows show the results of supervised 100% algorithm. Each box shows the violin plot of the results with respect to a given parameter. We show the results evaluated on full datasets and 1k datasets.

# J   LIGHTWEIGHT VERSUS HEAVYWEIGHT SEARCHES

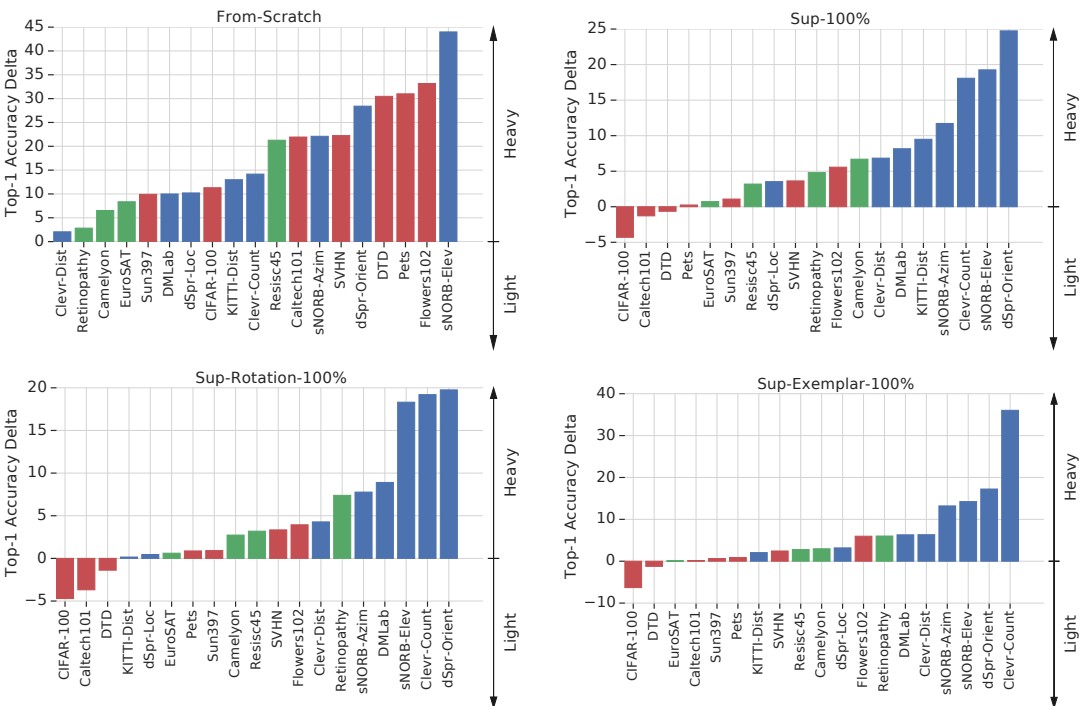

Figure 14: Per-dataset, absolute difference in top-1 accuracy of two hyper parameter sweep strategies. Top left: FROM-SCRATCH. Top right: SUP-100%. Bottom left: SUP-ROTATION-100%. Bottom right: SUP-EXEMPLAR-100% The bar colour denotes the task group as usual.

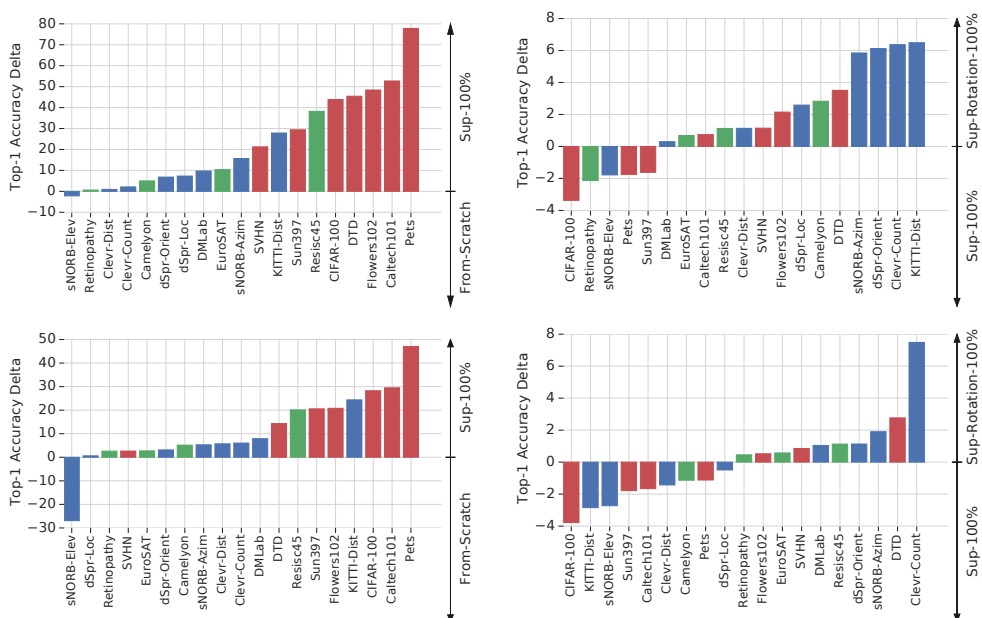

Figure 15: Absolute difference in top-1 accuracy between pairs of methods for each dataset. The bar colour denotes the task group as usual. Top: Lightweight hyperparameter sweep. Bottom: Heavyweight hyperparameter sweep. Left: SUP-100% versus FROM-SCRATCH – supervised pre-training yields a substantial improvement on the ● NATURAL datasets and some others. Right: SUP-ROTATION-100% versus SUP-100% – the additional self-supervised loss yields better representations for the ● STRUCTURED tasks.

# K  LINEAR EVALUATION

Here we describe the setup for linear evaluation. We follow  (Kolesnikov et al., 2019) and evaluate the frozen representation by training a linear logistic regression model. We use exactly the same hyperparameters as described in lightweight sweep from Section I. The only difference here is that only the linear layer is trained instead of fine tuning the whole network.

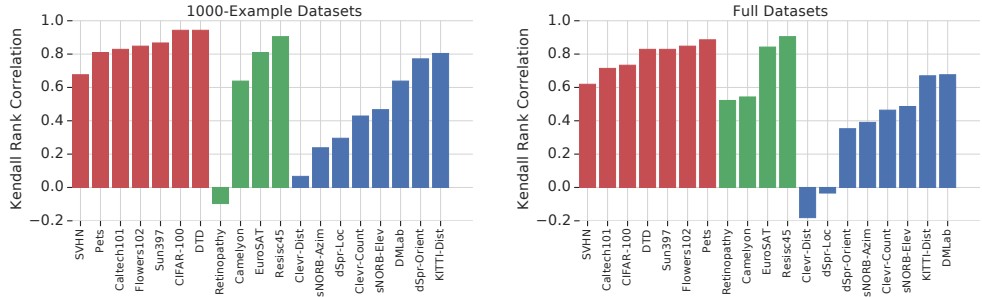

Figure 16: Kendall rank correlation coefficient between finetuning and linear evaluation on each dataset.

| | | Caltech101 | CIFAR-100 | DTD | Flowers102 | Pets | SVHN | Sun397 | Camelyon | EuroSAT | Resisc45 | Retinopathy | Clevr-Count | Clevr-Dist | DM-Lab | KITTI-Dist | dSpr-Loc | dSpr-Ori | sNORB-Azim | sNORB-Elev | Mean |
|---|---|---|---|---|---|---|---|---|---|---|---|---|---|---|---|---|---|---|---|---|---|
| | VAE | 34.9 | 8.0 | 7.8 | 10.1 | 7.3 | 23.8 | 3.4 | 63.8 | 38.4 | 13.2 | 73.6 | 26.9 | 52.3 | 22.6 | 38.3 | 60.7 | 11.1 | 19.3 | 20.8 | 28.2 |
| | WAE-UKL | 34.7 | 7.9 | 7.1 | 10.7 | 6.5 | 23.1 | 3.0 | 62.7 | 39.6 | 11.7 | 73.6 | 26.6 | 52.8 | 22.8 | 45.1 | 61.8 | 10.5 | 20.3 | 21.0 | 28.5 |
| | WAE-GAN | 32.8 | 7.5 | 6.2 | 11.3 | 6.9 | 25.5 | 3.2 | 65.1 | 43.0 | 11.0 | 73.6 | 28.0 | 53.4 | 23.9 | 45.4 | 60.0 | 10.2 | 19.6 | 22.7 | 28.9 |
| | WAE-MMD | 37.1 | 9.2 | 7.4 | 14.6 | 7.5 | 23.9 | 3.8 | 65.3 | 44.6 | 13.4 | 73.6 | 27.8 | 53.3 | 23.8 | 43.0 | 66.3 | 10.1 | 22.9 | 22.2 | 30.0 |
| | Un.C.-BigGAN | 51.8 | 14.3 | 29.5 | 38.6 | 11.7 | 37.9 | 8.8 | 78.2 | 68.7 | 29.7 | 73.4 | 34.6 | 48.5 | 25.4 | 30.1 | 30.1 | 24.0 | 26.6 | 28.5 | 36.3 |
| | Jigsaw | 58.6 | 14.4 | 39.0 | 41.8 | 18.3 | 40.8 | 10.8 | 77.1 | 85.9 | 49.7 | 73.5 | 39.3 | 49.3 | 28.4 | 50.8 | 32.3 | 20.3 | 26.3 | 28.4 | 41.3 |
| | Rel.Pat.Loc. | 62.9 | 15.4 | 43.5 | 49.1 | 20.6 | 41.6 | 12.7 | 74.9 | 86.0 | 54.2 | 73.4 | 39.4 | 49.1 | 28.0 | 53.0 | 29.8 | 19.4 | 26.9 | 28.8 | 42.6 |
| 1000 | Cond-BigGAN | 63.3 | 20.8 | 36.2 | 61.2 | 23.0 | 47.1 | 12.6 | 73.5 | 75.9 | 43.1 | 63.5 | 40.9 | 50.2 | 27.9 | 37.8 | 40.5 | 42.8 | 30.2 | 32.0 | 43.3 |
| | Exemplar | 57.8 | 20.4 | 40.7 | 56.6 | 27.3 | 42.8 | 13.1 | 78.6 | 88.3 | 57.6 | 74.1 | 45.2 | 50.9 | 28.1 | 59.3 | 55.1 | 28.7 | 25.6 | 33.1 | 46.5 |
| | Rotation | 66.0 | 22.8 | 44.1 | 48.4 | 17.4 | 53.9 | 14.4 | 78.3 | 87.2 | 50.3 | 73.7 | 42.0 | 53.5 | 30.9 | 61.9 | 59.7 | 30.3 | 26.5 | 37.8 | 47.3 |
| | Semi-Ex-10% | 82.3 | 40.4 | 55.3 | 75.1 | 82.8 | 48.8 | 27.5 | 77.9 | 89.0 | 64.7 | 68.1 | 35.6 | 39.6 | 28.9 | 65.8 | 29.6 | 38.2 | 26.5 | 26.5 | 52.8 |
| | Semi-Rot-10% | 81.8 | 38.7 | 56.8 | 76.8 | 81.6 | 40.5 | 27.2 | 79.9 | 91.1 | 70.1 | 66.9 | 39.2 | 39.4 | 28.1 | 62.7 | 45.0 | 31.6 | 21.8 | 27.6 | 53.0 |
| | Sup-Rot-100% | 86.8 | 50.8 | 65.4 | 86.3 | 89.5 | 42.6 | 35.7 | 79.3 | 92.3 | 76.9 | 68.9 | 40.5 | 37.4 | 32.1 | 68.5 | 39.7 | 34.0 | 25.5 | 26.7 | 56.8 |
| | Sup-100% | 87.7 | 50.9 | 64.7 | 83.5 | 89.3 | 50.7 | 35.7 | 81.5 | 91.6 | 72.9 | 70.5 | 41.0 | 40.0 | 29.0 | 65.0 | 42.6 | 38.4 | 27.1 | 26.8 | 57.3 |
| | Sup-Ex-100% | 87.8 | 50.9 | 64.0 | 84.5 | 89.1 | 53.3 | 35.4 | 78.4 | 92.0 | 73.3 | 68.5 | 39.6 | 40.0 | 31.1 | 72.0 | 42.1 | 39.5 | 27.7 | 27.6 | 57.7 |
| | VAE | 42.9 | 16.7 | 8.4 | 12.3 | 9.6 | 24.6 | 7.8 | 69.4 | 45.9 | 20.1 | 73.6 | 30.7 | 57.3 | 30.3 | 41.0 | 79.7 | 13.6 | 20.1 | 25.9 | 33.1 |
| | WAE-UKL | 42.2 | 17.3 | 7.8 | 13.1 | 9.1 | 24.8 | 8.2 | 67.5 | 49.7 | 21.3 | 73.6 | 31.0 | 59.5 | 31.1 | 45.3 | 82.0 | 13.9 | 21.0 | 26.2 | 33.9 |
| | WAE-GAN | 40.3 | 19.5 | 8.4 | 14.4 | 9.0 | 27.5 | 8.7 | 70.1 | 57.4 | 22.1 | 73.6 | 34.0 | 62.3 | 32.9 | 48.9 | 82.7 | 13.0 | 21.5 | 29.4 | 35.6 |
| | WAE-MMD | 43.3 | 19.4 | 8.6 | 17.4 | 9.3 | 26.2 | 9.5 | 71.0 | 55.0 | 23.6 | 73.6 | 33.0 | 62.1 | 31.9 | 50.6 | 85.5 | 13.7 | 22.1 | 27.7 | 36.0 |
| | Un.C.-BigGAN | 60.5 | 35.7 | 38.0 | 49.1 | 17.1 | 55.3 | 26.7 | 79.4 | 77.1 | 50.9 | 74.1 | 40.4 | 55.6 | 37.3 | 30.9 | 57.4 | 40.4 | 32.5 | 39.2 | 47.2 |
| | Jigsaw | 68.7 | 29.9 | 49.3 | 53.4 | 27.7 | 50.6 | 29.2 | 78.2 | 90.6 | 70.2 | 73.6 | 49.7 | 57.2 | 41.7 | 55.2 | 40.4 | 23.0 | 31.2 | 36.6 | 50.3 |
| | Rel.Pat.Loc. | 73.2 | 29.0 | 55.5 | 57.5 | 29.7 | 50.2 | 27.8 | 77.4 | 90.7 | 70.2 | 74.5 | 49.7 | 57.7 | 42.8 | 53.0 | 38.5 | 20.7 | 32.7 | 36.3 | 50.9 |
| Full | Exemplar | 68.2 | 49.2 | 52.2 | 67.4 | 38.3 | 59.3 | 43.7 | 81.5 | 94.2 | 79.9 | 74.8 | 56.3 | 58.8 | 41.4 | 62.8 | 70.6 | 34.2 | 32.3 | 45.0 | 58.4 |
| | Cond-BigGAN | 73.6 | 47.9 | 43.5 | 70.1 | 29.4 | 67.7 | 39.4 | 76.3 | 84.6 | 66.9 | 69.7 | 52.1 | 60.4 | 39.7 | 43.7 | 75.5 | 64.9 | 67.3 | 52.7 | 59.2 |
| | Rotation | 77.5 | 48.4 | 56.2 | 60.2 | 30.6 | 71.6 | 42.1 | 82.0 | 93.4 | 74.7 | 74.7 | 56.6 | 64.8 | 46.0 | 62.0 | 75.1 | 36.1 | 38.3 | 52.8 | 60.2 |
| | Semi-Rot-10% | 86.7 | 65.6 | 65.5 | 83.6 | 84.6 | 62.2 | 58.8 | 84.0 | 96.0 | 87.6 | 74.6 | 56.8 | 53.1 | 45.9 | 64.9 | 69.7 | 45.7 | 41.8 | 44.1 | 66.9 |
| | Semi-Ex-10% | 86.8 | 64.6 | 63.6 | 80.7 | 85.3 | 68.7 | 57.7 | 83.2 | 94.5 | 82.8 | 74.3 | 52.1 | 54.6 | 46.8 | 74.1 | 67.2 | 65.2 | 68.8 | 41.7 | 69.1 |
| | Sup-Rot-100% | 91.8 | 74.3 | 72.2 | 90.8 | 91.1 | 63.4 | 68.2 | 82.5 | 96.8 | 90.6 | 74.8 | 63.7 | 51.9 | 49.4 | 71.4 | 75.3 | 60.2 | 61.5 | 44.7 | 72.3 |
| | Sup-100% | 91.6 | 74.9 | 72.6 | 88.7 | 91.0 | 69.7 | 68.0 | 84.4 | 96.5 | 88.6 | 76.0 | 61.7 | 55.1 | 50.6 | 74.4 | 75.4 | 62.2 | 72.0 | 44.1 | 73.6 |
| | Sup-Ex-100% | 92.5 | 74.1 | 72.1 | 89.2 | 91.6 | 73.6 | 67.6 | 84.0 | 96.0 | 87.9 | 75.0 | 62.2 | 57.9 | 51.4 | 74.1 | 75.6 | 62.0 | 73.2 | 46.5 | 74.0 |

Table 9: Top-1 accuracy of all the models with linear evaluation on VTAB.

## L   SCALING UP THE ARCHITECTURE

In this section, we study the problem of "is scaling up the architectures helpful?". Figure 17 shows top-1 accuracy on ImageNet public validation set when scaling up the architectures. Widening factor is the multiplier on the network width, where ×1 stands for the standard ResNet architecture. Depth stands for the number of architecture layers. As expected, the model accuracy goes up with either wider or deeper architectures. Figure 18 shows the results on VTAB benchmark, where the 2x wider ResNet152 architecture performs consistently better than the standard ResNet50 model.

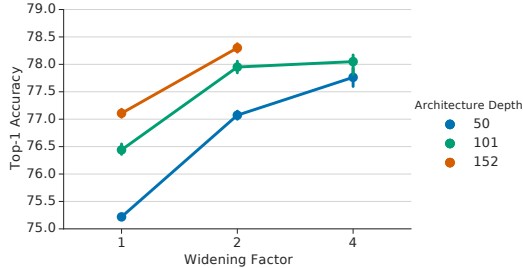

Figure 17: Top-1 accuracy on ImageNet public validation set when scaling up the architectures. The accuracy goes up with either wider or deeper architecture.

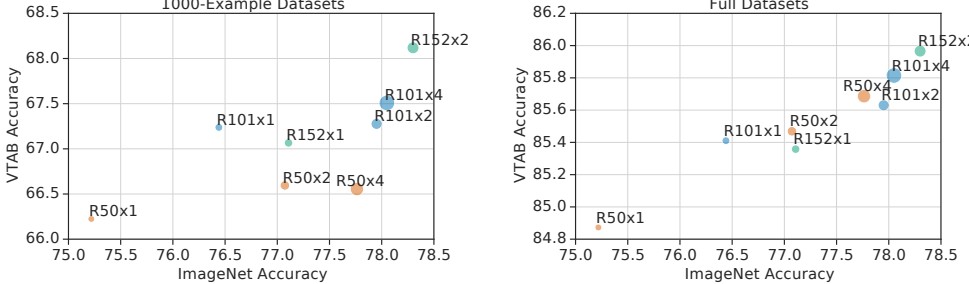

Figure 18: VTAB performance on the scaled up architectures. Here the architecture is represented by different color and width is represented by different circle size. The performance increases when switching from the standard ResNet50 architecture to ResNet152 2x wider architecture.

## M    BUDGET ANALYSIS

Figures 19 and 20 show the top-1 accuracy attained by SUP-100% and ROTATION using different schedule lengths when using 1000 samples or the full dataset, respectively.

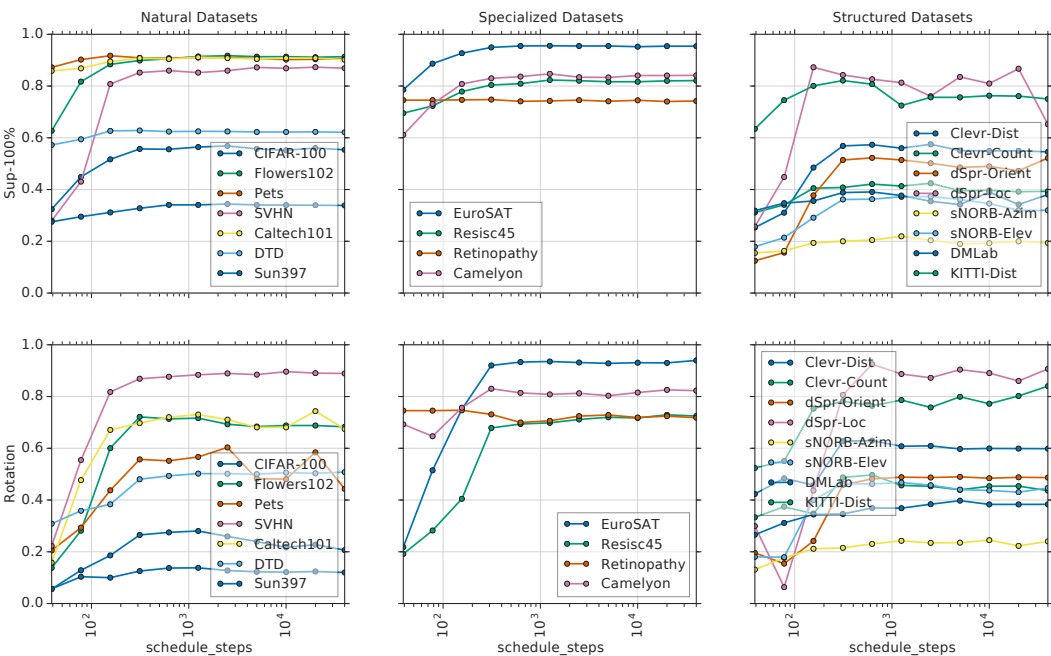

Figure 19: Top-1 accuracy for each task with respect to the number of fine-tuning steps on the 1000-example datasets. More steps is usually better, but performance is stable after 1000 steps.

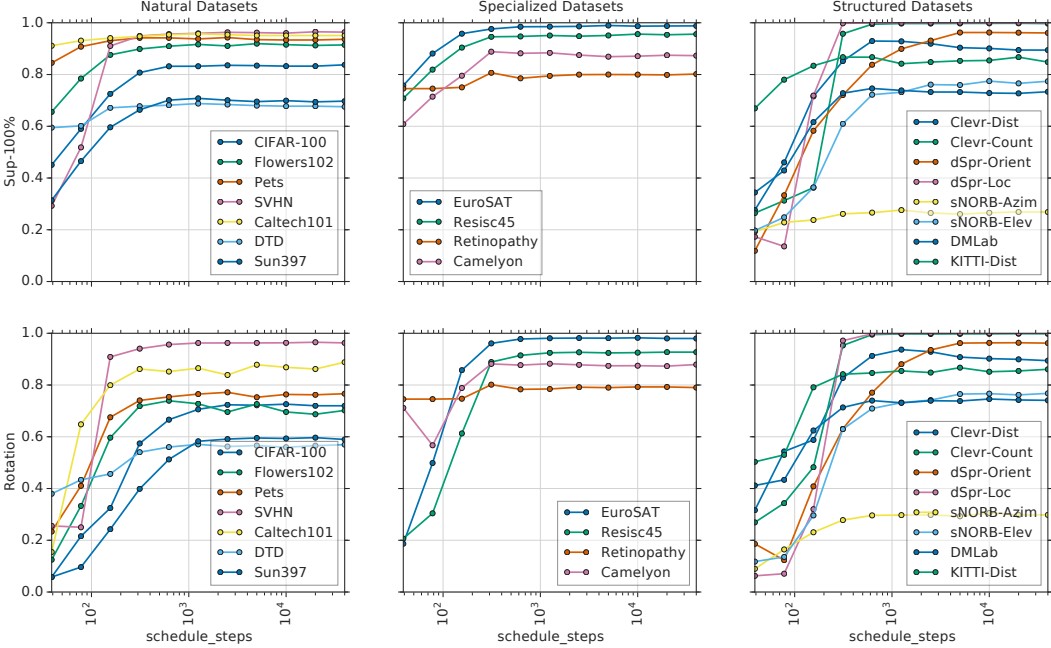

Figure 20: Top-1 accuracy for each task with respect to the number of fine-tuning steps on the full datasets. More steps is usually better, but performance is stable after 1000 steps.

# N    COMPARISON TO VISUAL DECATHLON

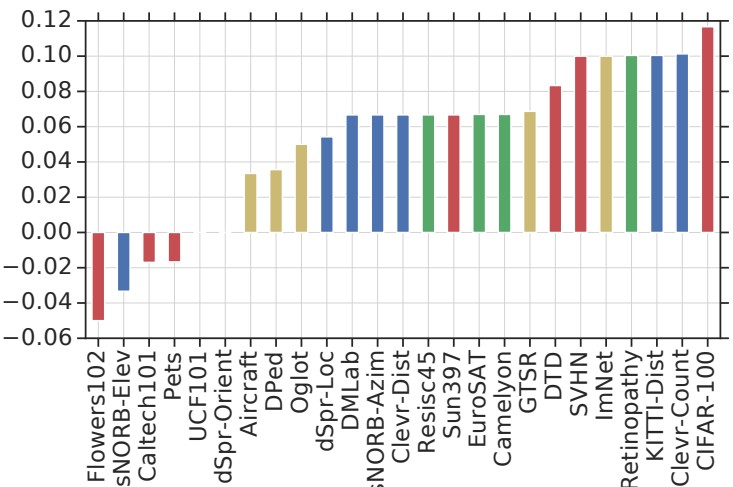

Figure 21: Absolute differences in Kendall's rank correlation score between the "gold" ranking in each dataset, and the ranking obtained with either VTAB or Visual Decathlon. Bar colors indicate the category as usual, and yellow indicates datasets only present in Visual Decathlon. Positive values indicate that VTAB ranking is closer than Visual Decathlon ranking, when compared against the "gold" ranking of a particular dataset. The average ranking correlation for VTAB is 0.76, and 0.70 for Visual Decathlon.

In Section 4 we describe the differences between VTAB and Visual Decathlon protocols. Here we answer a more practical question: which benchmark should one use to compare the adaptation abilities of a set of methods to unseen tasks? We will show that the rank correlation with an unseen task, is expected to be higher for the ranking obtained using VTAB, than using Visual Decathlon.

First, we fine-tuned each of our 16 baseline models in each of the Visual Decathlon datasets (using the lightweight hyperparameter search described in Section 3.1), which we downloaded directly from the competition's website. All datasets in the Visual Decathlon are provided so that the shorter size of the image is 72 pixels, while our baseline models were trained in much larger resolutions. To overcome this difference in resolution, we resize the Visual Decathlon images to the resolution required by each model. We checked that our baseline models obtained reasonable results on the Visual Decathlon benchmark. In fact, our best model, SUP-ROTATION-100%, reports a test average accuracy of 78.22%, and a decathlon score of 3580, which are both slightly better than the best results reported in Rebuffi et al. (2018) (78.08% and 3412, respectively), and other works (e.g. Rebuffi et al. (2017); Rosenfeld & Tsotsos (2018)). However, notice that comparing these numbers is delicate, since we did not use any data augmentation during training and all our models are based on the Resnet50 architecture, while these works use heavy data augmentation (that depends on the dataset), and Resnet26-like architectures.

Then, for each task $T$ in the union of VTAB and Visual Decathlon, we rank the 16 baseline methods according to their accuracy on task $T$. If we consider $T$ as the unseen dataset, this is the "gold" ranking of the studied methods. Now, we obtain two alternative rankings: one based on the mean accuracy in VTAB and another on Visual Decathlon, excluding task $T$ in both cases to avoid any bias. We can then compute, for each "unseen" dataset, the Kendall's ranking correlation between the "gold" ranking and each of the alternative rankings. Figure 21 shows the absolute differences in the rank correlation score between the VTAB-based ranking and Visual Decathlon-based ranking.

For most datasets, the difference is positive, which means that the ranking according to VTAB correlates better with the "gold", than the raking obtained using Visual Decathlon. Notice that even tasks that were not part of VTAB (colored in gold), are better represented by VTAB's ranking than that of Visual Decathlon. These results are not surprising, since VTAB contains a more representative set of tasks than Visual Decathlon. The average ranking correlation for VTAB is 0.76, and 0.70 for Visual Decathlon.

