# OpenReview forum: "The Visual Task Adaptation Benchmark"
_ICLR.cc/2020/Conference — Reject_

### Official Review · AnonReviewer3 · 2019-10-22
**Official Blind Review #3**

**Rating:** 8

**Review:**


###  Summary
1. The paper presents a new benchmark (VTAB) to evaluate different representation learning methods. VTAB mostly consists of a combination of existing classification datasets. (The classification tasks based on Clevr, SmallNorb, and DMLab are introduced by this paper.)
2. It evaluates and compares the existing representation learning method on the proposed benchmark with an extensive hyper-parameter search.

### Decision with reasons
I vote for accepting this paper (weak accept).

1. The experimental evaluation in this paper is outstanding. Different methods are compared fairly with extensive hyper-parameter tuning (Heavyweight tuning).

2. Even though I find the experiment work to be illuminating, I'm not fully convinced that we need more standardized benchmarks. I'm also not convinced by some of the choices made in designing the benchmark (Such as the models can be pre-trained on any data as long as there is no overlap with the evaluation data).

I find the experimental evaluation to be useful enough for considering accepting this paper, but I will not strongly advocate for acceptance (Unless the authors can better explain why we need yet another benchmark.)

### Supporting arguments for the reasons for the decision.

1. The paper compares existing representation learning methods in a very convincing way. Even though it misses some baselines (Meta-learning based representation learning), it compares supervised, self-supervised, semi-supervised, and generative methods comprehensively. Moreover, the hyper-parameter selection process for the benchmark is fully defined in a dataset agnostic way (It does not depend on the evaluation dataset). This is more important than it might seem -- fully specifying algorithms is important and currently rarely done in the research literature.

I'm not aware of any existing work that compares various representation learning methods on a diverse benchmark fairly and comprehensively, and I'm sure I'll be using the results in this paper to support my claims in the future. As a result, I think this work is worth accepting just for its thorough evaluation of existing methods.

2. However, the paper does not add much to discourse apart from the thorough evaluation. Benchmarks are already prevalent in representation learning, and VTAB doesn't focus on a single well-defined problem with-in representation learning.

For example, dSprites is a benchmark to evaluate how well algorithms can disentangle the underlying data generation factors. An algorithm that significantly improves results on dSprites would, as a result, be better at learning disentangled representation. VTAB, on the other hand, is not trying to highlight any specific problem in representation. As a result, it's not clear what it means if an algorithm does better on VTAB (Apart from the vague notion that it is learning 'better' representations.)

This problem is especially exacerbated by the fact that VTAB does not restrict representation learning to a fixed dataset. This means that a trivial way of improving performance is to just get more representative data (Which is not a bad way of improving performance, but not the way that requires scientific study).

### Questions for Authors

1. The paper makes a binary distinction between fine-tuning and linear evaluation when in-fact this is more nuanced. I suspect that the best performance would be achieved by feezing layers up to a point (And not just not freezing any layers or freezing everything and training a linear classifier). Did the authors explore other choices, and if not, why do they think the current binary distinction is enough. (From my perspective, it makes sense to freeze the initial layers which might be extracting more local/general features and adapt the later layers. The boundary point should be a hyper-parameter)

2. Currently, VTAB puts no restrictions on the representation learning dataset as long as the representation learning dataset does not over-lap with the evaluation dataset. This is problematic since an easier way of improving results on VTAB is to just collect more data/ data that is more representative of evaluation tasks. One could argue that the point of VTAB is to not use details of evaluation datasets in making any decisions for representation learning, however, it is hard to enforce this. For example, if I augment Imagenet with a medical dataset different from the ones in VTAB, is that fine or does that count as cheating?

I feel that a much better way of defining the benchmark would be to restrict representation learning to Imagenet.

### Other minor comments
I don't have many minor comments since the paper is very clear and well written. The author could consider augmenting their graphs with patterns in addition to color to make paper friendly for grey-scale printing and use a more colorblind-friendly color combination.


### Post discussion update ###

One of my primary concerns was the benchmark didn't restrict representation learning to a fixed dataset. That made it trivial to improve performance: just train on a more representative dataset. However, as authors clarified, any dataset collected conditioned on the test set is against the spirit of VTAB. Moreover, discovering what kind of data improves representations is also a research problem, and by restricting to a fixed pre-training dataset, VTAB would not be suitable for such research directions.

I agree that solving representation learning might not just be about finding a better algorithm, but a better algorithm coupled with the right data. In that sense, VTAB's decision to allow any data for pretraining makes sense. However, I'm also concerned that the benchmark might push people to (directly or indirectly) start training on data more representative of VTAB evaluation tasks, and the performance gains in the benchmark will not translate to better algorithms. However, I don't see how this can be avoided for an open-ended benchmark. Since the authors have agreed to include guidelines on how to use the benchmark without abusing it, letting the community police itself given the guidelines is not a bad idea.

I can see VTAB as a benchmark for answering interesting questions (mentioned by authors in the response) such as:

1- How to incorporate large amounts of unlabelled data for representation learning. For example, can we learn better representations by self-supervised learning on a large corpus of data (Such as unlabelled youtube videos)
2- Can we generate data using simulators that can help us learn better representations?

Finally, the authors agreed to add meta-data in their leaderboards (which they plan to release after the review period). The meta-data would allow us to compare methods at a more granular level, such as comparing different unsupervised learning methods. Overall, I think the benchmark and the leaderboards would act as a good source for comparing different unsupervised, semi-supervised, and supervised learning methods (and their combination) fairly.

I'm also updating my score from a weak-acceptance to an acceptance.

**Experience Assessment:**

I have published one or two papers in this area.

**Review Assessment: Checking Correctness Of Derivations And Theory:**

N/A

**Review Assessment: Checking Correctness Of Experiments:**

I carefully checked the experiments.

**Review Assessment: Thoroughness In Paper Reading:**

I read the paper thoroughly.

---

> ### Author Response · Authors · 2019-11-07
> **Thanks for highlighting some important points**
>
> Thanks for your thorough feedback. You raise a number of important points. We have considered the design choices for VTAB carefully, and we hope that we can convince you during discussion that these are appropriate for this benchmark.
>
>
> ### Permitting any upstream data
>
> VTAB was designed to measure progress towards universal visual representations: those that reduce sample complexity on any classification task a human can perform. Insisting that all methods pre-train on ImageNet would artificially limit progress.
>
> Discovering how to exploit other datasets for this goal is valuable research, and there are many options e.g. generating data (using rendering or generative model), mining large amounts of weakly supervised data, training on videos, etc. We intend that VTAB measures such progress. Analogously, the GLUE benchmark (gluebenchmark.com) does not restrict pre-training data, and has driven significant progress in NLP.
>
> Indeed, for scientific study, confounding variables should be controlled. In the experiments we present here, we study the upstream training method. We therefore control for dataset, architecture, transfer strategy, etc. Other methods that also train upstream on ImageNet may compare in-class. However, these variables also influence representation quality, and VTAB may be used to measure progress along these axes also.
>
>
> ### Collecting data conditioned on the test tasks
>
> The only constraint is that the test tasks are considered unseen. Similarly, the test sets for individual tasks should be unseen. The straw-man strategy proposed: collecting data conditioned on the downstream tasks, would violate the VTAB protocol.
>
> The popular ImageNet benchmark also has a leaderboard for methods that use additional data (https://paperswithcode.com/sota/image-classification-on-imagenet). However, it is understood that such data collection is not conditioned on the ImageNet test set. As with most long-running scientific benchmarks, the community must police itself in distinguishing methods that follow the spirit of the benchmark, and those that do not.
>
>
> ### Alternative Protocols
>
> We discuss in the paper many alternative evaluations --- linear eval, Facebook AI SSL, meta-dataset, Visual Decathlon --- and make empirical comparisons where appropriate. For reasons presented in the paper, none of these satisfy the goal of VTAB: to measure end-to-end performance with limited data on many tasks, beyond natural image object type classification.
>
> As you note, protocols exist to measure specific aspects (e.g. dSprites for disentanglement). These are also valuable. However, VTAB fulfils a different role to measure end-to-end performance and cross-task generalization (like GLUE in NLP).
>
> Do you have a particular benchmark in mind that achieves the same objectives as VTAB?
>
>
> ### Linear Evaluation and Fine-tuning
>
> We do not intend to make a hard distinction between linear evaluation and fine-tuning. Any transfer strategy is permitted in VTAB. In the experiments we present, to compare upstream training algorithms, we fix other variables. We use fine-tuning as a default transfer strategy because it usually performs best.
>
> We have looked into partial freezing, but full fine-tuning worked better for us. We hope that future research can improve upon fine-tuning.
>
> We believe that the linear transfer constraint is unnecessary, if the goal is sample efficiency. We make an explicit comparison to this strategy only because that is a popular protocol in the literature.
>
>
> ### Additional Baselines
>
> As discussed below with Abhishek Sinha, we agree that evaluating other methods, such as meta-learning, is interesting. We hope that you agree that the study we present is already very large for one paper, and we could not include everything. We selected methods based on availability and reproducibility. The goal of this paper, and the released code, is to provide the opportunity compare of the vast number of possibilities on a challenging and fair benchmark.

---

> > ### Comment · AnonReviewer3 · 2019-11-12
> > **Re: Thanks for highlighting some important points**
> >
> > Thank you for the clarification.
> >
> > I have read the response in detail, and it has clarified some of my main concerns. I will update my review in the coming days to reflect this (hopefully before the rebuttal period ends to give the authors time to respond to my update).
> >
> > Although I agree that letting the community police itself in using the benchmark properly is reasonable, I think as proposers of a new benchmark, it's the responsibility of authors to make any potential abuse as unlikely as possible. Being explicit about some possible ways of abusing the benchmark would be one way of achieving this. Note that benchmarks are abused in our community all the time.
> >
> > For example, if a transfer algorithm has orders of magnitude more hyper-parameters and uses orders of magnitude more compute budget to tune these parameters compared to the fine-tuning baseline, is that a fair comparison? Shouldn't the fine-tuning be given equal compute budget to better tune its parameters too? I'm sure the authors can think of more cases of unfair comparisons which they can explicitly mention in their paper.
> >
> > ## An unrelated question
> > Since VTAB parallels GLUE in many ways, I was wondering if authors were also planning to host public leaderboards similar to GLUE. I certainly don't think this is necessary as long as the authors release code that allows anyone to use the benchmark easily (Which they have already attached with this submission), but it'd still be a nice addition.

---

> > > ### Author Response · Authors · 2019-11-12
> > > **Re: Thanks for highlighting some important points**
> > >
> > > Thanks for following up. We shall add a discussion paragraph highlighting the need for fair comparison when using the benchmark. We designed the lightweight evaluation mode to make explicit the importance of fair hyperparameter selection, and to attain meaningful results without a large computational budget.
> > >
> > > To summarize, the only true abuse of the benchmark would be to condition on the test tasks. Examples of other factors that should be controlled for, or whose optimization with respect to VTAB should be made explicit, are:
> > >
> > > - Hyperparameter sweep size
> > > - Hyperparameter search algorithm
> > > - Architecture
> > > - Computational budget
> > > - Downstream image preprocessing
> > > - Transfer algorithm
> > > - Upstream training data
> > >
> > > We have a public leaderboard ready to reveal when anonymity is lifted. We will provide functionality to add metadata, so that the above factors can be considered when comparing methods.

---

> > > > ### Comment · AnonReviewer3 · 2019-11-13
> > > > **Review updated**
> > > >
> > > > I've updated my original review reflecting the discussion (See "### Post discussion update ###" appended to the original review) and changed my score to an acceptance.
> > > >
> > > > I did not address "### Linear Evaluation and Fine-tuning" and "### Additional Baselines" in the update as I agree that incorporating everything in one paper is difficult, and the paper already does quite a lot.

---

### Official Review · AnonReviewer2 · 2019-10-24
**Official Blind Review #2**

**Rating:** 6

**Review:**


The paper essentially addresses the difficult problem of visual representation evaluation. It attempts to find a universal way of assessing the quality of a model's representation. The authors define a good representation as one that can adapt to unseen tasks with few examples. With this in mind, they propose Visual Task Adaptation Benchmark (VTAB), a benchmark that is focused on sample complexity and task diversity.

- Very clear, well written and well structured. (Although not fully self contained in the main body of the paper - 20 pages of supplementary material!)
- The benchmark tasks are constrained to unseen tasks, which seems obvious but is often violated when evaluating representations
- It does a good attempt at covering a large spectrum of realistic domains (19 tasks!) to assess generality.
- Extensive study is conducted, covering the published state of the art methods in each domain.
- The study leads to interesting finding, such as promising results on self-supervision and negative results on generation.

Overall, I believe the paper is an important contribution as it provides some interesting analysis of the current state of the art for visual representation learning.

**Experience Assessment:**

I do not know much about this area.

**Review Assessment: Checking Correctness Of Derivations And Theory:**

N/A

**Review Assessment: Checking Correctness Of Experiments:**

I assessed the sensibility of the experiments.

**Review Assessment: Thoroughness In Paper Reading:**

I made a quick assessment of this paper.

---

> ### Author Response · Authors · 2019-11-07
> **Thank you for the positive comments**
>
> Thank you for your very positive comments. Do you have feedback on any aspects that could improve this work?
>
> We recognise that we defer many details and analyses to the Appendix. It was challenging with both a new benchmark and an extensive study, to distill the most important points into 10 pages. If you think that there are aspects that we should re-prioritize, we would be happy to try to refactor.

---

### Official Review · AnonReviewer1 · 2019-10-28
**Official Blind Review #1**

**Rating:** 3

**Review:**

This paper presents a Visual Task Adaptation Benchmark (VTAB) as a diverse and
challenging benchmark to evaluate the learned representations. The VTAB judges whether the learned representation is good or not by adapting it to unseen tasks which have few examples. This paper conducts popular algorithms on a large amount of VTAB studies by answering questions: (1) how effective are ImageNet representation on non-standard datasets? (2) are generative models competitive? (3) is self-supervision useful if one already has labels?

What is the size of the training dataset in the source domain?

The authors need to compare the conclusion obtained with other works. It seems that there is no new founding in this paper.

**Experience Assessment:**

I have published in this field for several years.

**Review Assessment: Checking Correctness Of Derivations And Theory:**

I assessed the sensibility of the derivations and theory.

**Review Assessment: Checking Correctness Of Experiments:**

I assessed the sensibility of the experiments.

**Review Assessment: Thoroughness In Paper Reading:**

I read the paper at least twice and used my best judgement in assessing the paper.

---

> ### Author Response · Authors · 2019-11-07
> **Could you please be more specific**
>
> In our experiments, we train all networks upstream on the standard ILSVRC2012-ImageNet dataset which contains 1.28M images and 1000 classes.
>
> We motivated our new benchmark in depth, and provided a very large scale analysis of much of the current field. Could you please be more specific about the aspects that you believe are missing?

---

### Public Comment · ~Abhishek_Sinha1 · 2019-10-17
**Experiments on BigBiGAN and other self-supervision techniques.**

Hi, the paper mentions that the generative models fail to learn a good representation as shown by their poor performance over the 19 tasks. Did the authors also experiment with using BigBiGAN for representation learning? BigBiGAN has been shown to achieve a good linear predictor accuracy over the ImageNet dataset.

Also, did the authors also try other self-supervision techniques, apart from rotation and exemplar, like CPC, CPC++ or CMC for their experiments?

---

> ### Author Response · Authors · 2019-10-22
> **We selected representative methods for this paper, and hope the community will use VTAB for concurrent & future work.**
>
> Hi Abhishek,
>
> Thanks for the interest! We selected a representative subset of algorithms from the various methods classes: generative, self-supervised, etc. (of course, we could not implement everything!) We chose methods that are 1) Published 2) Well established, state-of-the-art, or near state-of-the-art in their respective domains. 3) Have released code or a TensorFlow Hub model, or are relatively straightforward to reproduce faithfully.
>
> BigBiGAN, for example, would be very interesting to be evaluated with VTAB, but since the Hub model was released only just before the paper submission deadline, it is not included here. Similarly for other concurrent work.
>
> The primary goal of our paper is to release a comprehensive evaluation protocol. We will continuously update the results with new methods, and make them public (at a URL to be disclosed after the review period); however, this alone will not scale as new representation learning algorithms are invented. We hope the community will use the code we release to benchmark all of the methods that they care about.

---

### Decision · Program_Chairs · 2019-12-19

**Decision:**

Reject

**Comment:**

The authors present a new benchmark for evaluating a plethora of models on a variety of tasks. In terms of scores, the paper received a borderline rating, with two reviews being rather superficial unfortunately. The last reviewer was positive. The reviewers generally agreed that the benchmark is interesting and carries value, and the AC agrees. Authors certainly invested a significant effort in designing the benchmark and performing a detailed analysis over several tasks and methods. However, the effort seems more engineering in nature and insights are somewhat lacking. For an experimental paper, presenting the results is interesting yet not sufficient. A much more in-depth analysis and discuss would warrant a deeper understanding of the results and open directions for future work. This part is currently underwhelming.